# Bimodal ionic photomemristor based on a high-temperature oxide superconductor/semiconductor junction

Ralph El Hage[1], Vincent Humbert [1], Victor Rouco [1], Gabriel Sánchez-Santolino [2], Aurelien Lagarrigue[1], Kevin Seurre [1], Santiago J. Carreira[1], Anke Sander [1], Jérôme Charliac [3], Salvatore Mesoraca[1], Juan Trastoy[1], Javier Briatico [1], Jacobo Santamaría[1,2] & Javier E. Villegas [1] ✉

Memristors, a cornerstone for neuromorphic electronics, respond to the history of electrical stimuli by varying their electrical resistance across a continuum of states. Much effort has been recently devoted to developing an analogous response to optical excitation. Here we realize a novel tunnelling photo-memristor whose behaviour is bimodal: its resistance is determined by the dual electrical-optical history. This is obtained in a device of ultimate simplicity: an interface between a high-temperature superconductor and a transparent semiconductor. The exploited mechanism is a reversible nanoscale redox reaction between both materials, whose oxygen content determines the electron tunnelling rate across their interface. The redox reaction is optically driven via an interplay between electrochemistry, photovoltaic effects and photo-assisted ion migration. Besides their fundamental interest, the unveiled electro-optic memory effects have considerable technological potential. Especially in combination with high-temperature superconductivity which, in addition to facilitating low-dissipation connectivity, brings photo-memristive effects to the realm of superconducting electronics.

The search for faster, energy-efficient memories and novel computation schemes has fostered the exploration of resistive switching effects[1]. Observed in a variety of systems that span from magnetic[2,3] or ferroelectric[4,5] tunnel junctions to transition-oxide capacitors[6,7] and strongly correlated materials[8–10], the term resistive switching denotes a "jump" between non-volatile electrical resistance states (high/low resistance, 0/1 for logics) generally triggered by a voltage or a current pulse.

Memristors[11] constitute a particular class of two-terminal resistive switching devices whose functionality is beyond that of a binary memory: they show a continuum of states - instead of only two - and the switching between them is driven by the time-integrated current across the device (or by the history of applied voltages). Thus, one can think of memristors as multi-state memories switchable by cumulative stimuli. Depending on the resistance states' lifetime and dynamics, memristors can mimic the function of either synapses or neurons[11,12], thus constituting a cornerstone for the nascent field of neuromorphic computing[13–16].

A related, tantalizing idea is the development of memristors sensitive to light[17–25] –particularly, in which illumination can trigger a switching across non-volatile resistance states. This type of optical memory may be game-changing in applications such as photonic neural networks[26] and open the door to novel optoelectronic devices[27,28] –for example, neuromorphic vision sensors[21,29]. Various systems have been recently explored in search of photomemristive effects. Those include complex nanostructures based on optically

[1]Unité Mixte de Physique, CNRS, Thales, Université Paris-Saclay, 91767 Palaiseau, France. [2]GFMC, Dpto. Física de Materiales. Universidad de Ciencias Físicas, Universidad Complutense de Madrid, 28040 Madrid, Spain. [3]Laboratoire de Physique des Interfaces et des Couches Minces (UMR7647), CNRS, Ecole Polytechnique, 91128 Palaiseau Cedex, France. ✉e-mail: javier.villegas@cnrs-thales.fr

active polymers[17], metal-oxide capacitors[21,22,24], all-oxide[19] and semiconductor[18] heterostructures, as well as ferroelectric tunnel junctions[23] among others[30,31]. Their conductance shows photosensitivity due to mechanisms that span from light-induced polymer contraction/expansion[17] and electron trapping/detrapping[18,19,22,24] to photoinduced ferroelectric switching[23]. In those systems, the resistance states' lifetimes range from typically seconds in electronic processes[18,19,21,22,24] to minutes in organic systems[17], up to the virtual non-volatility of ferroelectric devices[23].

In addition to the common challenges associated with memristors, e.g. obtaining large resistance variations to facilitate readout and finding geometries enabling the high connectivity required for neuromorphic circuits[11,14–16], photo-memristors pose additional specific ones. Namely, in memristors, electrical excitation often allows for bidirectional switching –the polarity of the electrical stimulus determines whether the resistance increases or decreases. Such property is crucial for mimicking the so-called depression and potentiation phenomena characteristic of synaptic plasticity[14,15,17,18]. An analogous function is generally absent in the optical response (in most cases illumination only produces a resistance decrease[19,21–23]) except for a few realizations that require the combination of various light sources. For instance, in polymer-based devices the opposite effects of circularly and linearly polarized light respectively lead to a resistance increase or decrease, thus mimicking depression and potentiation[17]. Illumination under variable wavelength has also been exploited to that end, particularly in systems based on electron trapping/detrapping[24], in which the natural relaxation of visible-light excited conductance states is potentiated via infrared illumination, thus allowing for a virtually bidirectional optical switching.

Here we report on a new class of photo-memristor that exploits a distinct microscopic mechanism: a controllable oxygen exchange between the two materials that constitute the device –a superconducting cuprate and a semiconducting oxide. Crucially, such a mechanism allows for giant resistance-switching effects that can be driven both optically and electrically. Indeed, a remarkable specificity of our ionic photo-memristor is that its response to a given optical stimuli depends on the electrical history. Due to this entanglement, and at variance with other approaches, a single light source can controllably produce bidirectional switching. This behaviour results from an unusual, competing interplay between electrochemistry, photon-activated oxygen diffusion and strong photovoltaic effects. Interestingly, the present demonstration is based on a high-temperature superconducting cuprate. As further discussed below, although superconductivity is not a necessary ingredient for the photo-memristive effects, it greatly multiplies their technological potential,

not only because it enhances resistive switching effects and facilitates the connectivity required for neuromorphic circuits, but also because electro-optical memory is a novel, game-changing function in the thriving field of superconducting electronics[32–37].

The key specificities of the electro-optical switching behaviour observed here are schematically summarized in Fig. 1. The application of voltage pulses $V_{write}$ (of the order of a few Volts) enables a conductance switching $\triangle G_E$ across a continuum of levels, whose non-destructive readout is possible with a much lower $V_{read}$ of the order of mV. The conductance levels span between two extreme states –hereafter called ON (high conductance) and OFF (low conductance) – that can be up to four orders of magnitude afar. The electrical switching is hysteretic, bidirectional, and reversible. The ON state and intermediate resistance levels are metastable, and their lifetime depends on temperature: they are virtually non-volatile at low $T$ (tens of K), and slowly relax into the OFF state at higher $T$, at a rate that increases with temperature. Overall, this behaviour is reminiscent (*mutatis mutandis*) of the tunnelling electroresistance (TER) of ferroelectric tunnel junctions[23,38–40]. Strikingly, illumination with visible or UV light also leads to conductance switching, and it does so in a very characteristic fashion: the conductance level shifts in opposite directions depending on the previous electrical junction's state. Namely, optical stimuli lead to either an enhancement or a decrease in the conductance $\triangle G_{Op}$ depending on whether the device had been previously set in (or nearby) the OFF or ON state. While the electric history determines the sign of $\triangle G_{Op}$, crucially the amplitude of $\triangle G_{Op}$ cumulatively depends on the number of photons shone on the device, that is, it is controlled by the optical history. In addition to the electro-optical switching summarized in Fig. 1, the devices studied here show unusually large photovoltaic effects which, as discussed thereafter, play a key role in the photo-memristive behaviour.

## Junction's fabrication and interface characterization

The scheme of the photo-memristor is shown in Fig. 2a. It consists of a micrometric junction between the archetypal high-$T_C$ superconductor $YBa_2Cu_3O_{7−\delta}$ (YBCO, bottom electrode) and the transparent semiconductor indium-tin-oxide (ITO, top electrode). The junctions are fabricated on 30 nm thick c-axis oriented YBCO films grown epitaxially on (001) $SrTiO_3$ (STO) substrates. A ~ 1 μm-thick photoresist is spin-coated on the YBCO film and micrometric openings are photolithographed. The ITO subsequently deposited contacts the YBCO surface across the resist openings, forming the junctions (for further details see[41] and Methods). The as-grown YBCO/ITO interface was characterized by Scanning Transmission Electron Microscopy (STEM) and Electron Energy Loss Spectroscopy (EELS), displayed in Fig. 2b–d. The atomic resolution Z-contrast STEM image (Fig. 2b) demonstrates highly epitaxial YBCO growth on the $SrTiO_3$ (001) substrate. The ITO layer displays an amorphous matrix with nanocrystalline regions of a few tens of nm in size and different crystalline orientations. Evidence of oxygen exchange between both materials is provided by the EELS analysis. On the one hand, the Cu $L_{2,3}$ edge shows a chemical shift towards higher energies at the interface with ITO (Fig. 2c), which indicates[42,43] a local decrease of the Cu oxidation state (electron doping). On the other hand, and consistently, the analysis of the oxygen $K$ edge (Fig. 2d) reveals that the pre-peak feature (highlighted by the Gaussian fits) decreases in intensity when moving towards the interface, indicating a reduced hole carrier density[44]. In summary, the EELS analysis of both the Cu $L_{2,3}$ and O $K$ edges indicate electron doping of the interfacial YBCO, which is consistent with local oxygen depletion. That is as we observed earlier in YBCO/$Mo_{80}Si_{20}$ junctions[41], and is explained by the high reduction potential of copper in YBCO, $E_0(Cu) = 2.4$ V, as compared to that of indium in ITO, $E_0(Cu) = −0.49$ V (Supplementary Table 1). From this, a spontaneous reduction of the interfacial YBCO is expected upon deposition of ITO. Based on

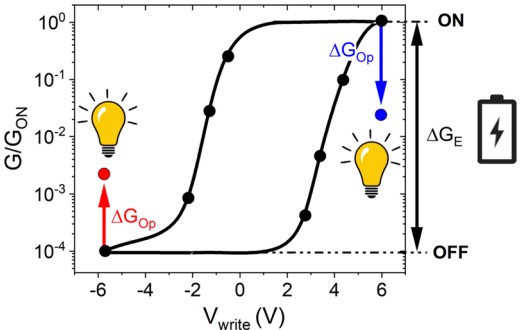

**Fig. 1 | Dual optical-electrical conductance switching.** Sketch and experimental data of a typical hysteresis loop showing the differential conductance measured on YBCO/ITO tunnel junctions as a function of the writing pulse voltage. The black arrow indicates the maximum amplitude of the electrical switching $\triangle G_E$. The red and blue arrows respectively indicate the amplitude of the optical switching $\triangle G_{Op}$ in the OFF and ON states respectively. Optical stimuli drive the junction towards intermediate states.

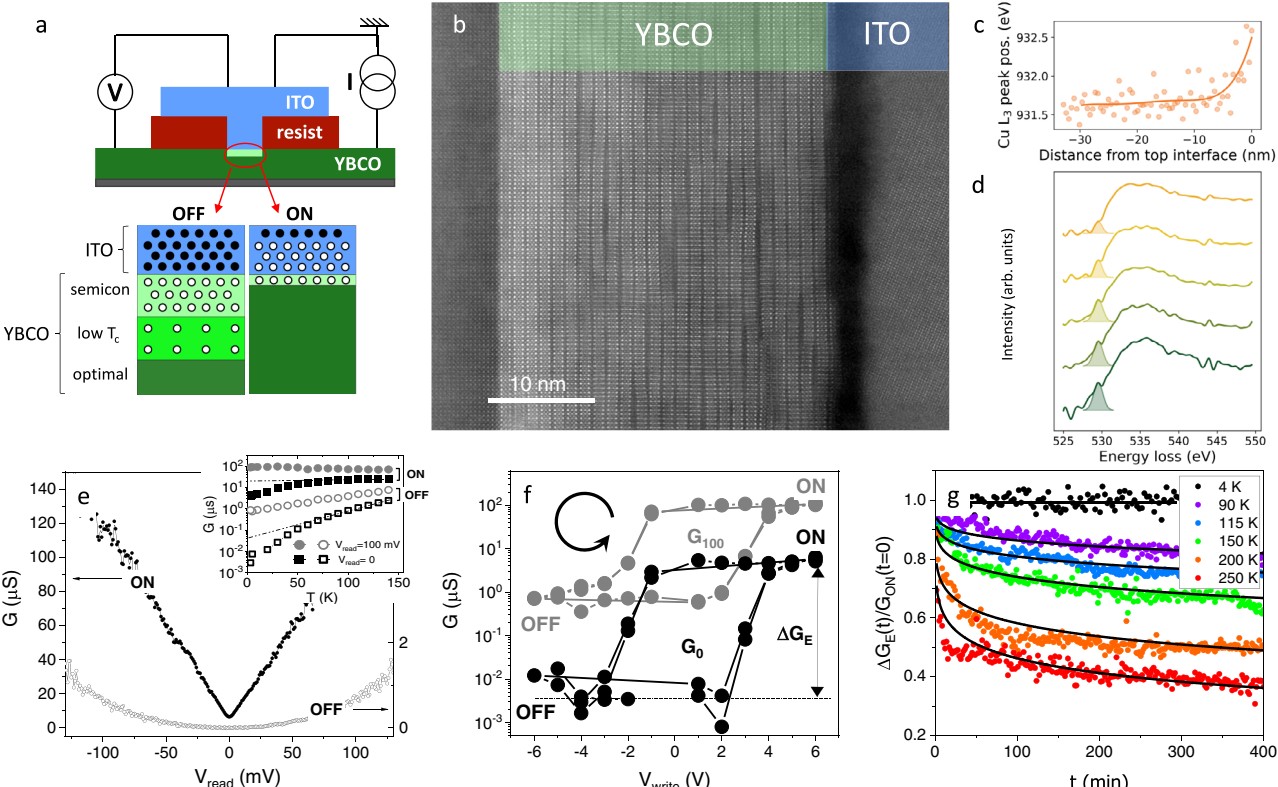

**Fig. 2 | Junction's structure and electrical switching. a** Scheme of the photo-memristor composed of a c-axis $YBa_2Cu_3O_{7-\delta}$/ITO tunnel junction grown on $SrTiO_3$(001). The micrometric junction is defined by depositing ITO into an opening through the insulating photoresist. Sketches of the YBCO/ITO interface in the OFF and ON states are displayed below. Solid and hollow circles respectively represent oxygen atoms and vacancies. **b** Atomic resolution Z-contrast STEM image of a YBCO/ITO bilayer grown on STO. **c** Cu $L_3$ peak position along the 30 nm YBCO thin film shows a chemical shift towards higher energies at the ITO interface. **d** Electron energy loss spectra at the O $K$ edge along the YBCO thin film. Gaussian fits of the oxygen pre-peak are also shown. The EELS intensity is normalized and shifted vertically to show the peak variation along the line scan. The different spectra are obtained from a line scan and correspond to 6 nm averages of the O $K$ edge centred at (from top to bottom) 3, 9, 15, 21 and 27 nm from the interface. **e** Differential conductance $G \equiv dI/dV_{read}$ as a function of $V_{read}$ after applying a

voltage pulse $V_{write}(\sim V)$ at T = 3.2 K to access the ON ($V_{write} > 0$) and OFF ($V_{write} < 0$) states. The inset displays the temperature dependence of the differential conductance at zero-bias $G_0$(squares) and $V_{read} = 100$ mV at $G_{100}$ (circles) in the ON (solid) and OFF (hollow) states. At zero bias ($G_0$, squares) the conductance drops at a higher pace below a certain temperature which is close to the superconducting transition. **f** Hysteretic behaviour of the differential conductance as a function of the writing voltage $V_{write}$ for $V_{read} = 100$ mV ($G_{100}$, grey, left scale) and for $V_{read} = 0$ ($G_0$, black, right scale), showing the maximum electrical switching amplitude $\triangle G_E$. The junction was cycled twice from positive to negative voltages as shown by the spinning arrow. **g** Relaxation of the normalized conductance switching $\Delta G_E(t)/G_{on}(t = 0)$ as a function of time after the junction is set in the ON state ($V_{write} = 6$ V at T = 3.2 K) measured at different temperatures (see legend), along with the best fits to a stretched exponential as discussed in Supplementary Fig. S4.

the profiles in Fig. 2c, the oxygen depletion in YBCO extends over $\sim$ 5-6 nm from the interface. According to earlier experiments[41], the doping gradient observed in Fig. 2c yields a gradient of physical properties as sketched in Fig. 2a (under the "OFF" label), having insulating YBCO at the interface, followed by YBCO with depressed superconducting properties (low $T_c$ YBCO) and by optimal oxygenation (and $T_C$) farther from the interface. As we discuss in the manuscript, the conductance switching triggered by applying $V_{write}$ or by illumination reflect changes in the oxygen content at both sides of the YBCO/ITO interface.

## Results
### Electrical switching
Figure 2e shows the typical differential conductance $G \equiv dI/dV_{read}$ vs. $V_{read}$ of YBCO/ITO junctions in the ON and OFF states, measured using the electrical wiring sketched in Fig. 2a. The bias dependence of the conductance is similar to that of YBCO/$Mo_{80}Si_{20}$[41] junctions, for which we demonstrated that, at low $V_{read}$, the dominant conduction mechanism is electron tunnelling. This is evidenced by two hallmarks: (i) an OFF-state parabolic $G(V_{read})$ consistent with the Brinkmann-Dynes-Rowell (BDR) model[45] for electron tunnelling (see Supplementary Fig. S1); and (ii) a strong suppression of the conductance around

zero bias as the temperature is decreased below the YBCO's $T_C$ (see Supplementary Fig. S2), which is as expected for electron tunnelling into a superconductor[46].

The conductance level in the ON state is orders of magnitude above the OFF state (notice that the ON/OFF curves in Fig. 2e respectively refer to the left/right y-axis). The junction can be reversibly switched from ON to OFF and vice versa, crossing over a continuum of intermediate states, by applying negative/positive $V_{write}$ (in the few volts range). This is shown in Fig. 2f, which displays the remnant conductance at $T = 4$ K for $V_{read} = 100$ mV and $V_{read} = 0$ mV (labelled $G_{100}$ and $G_0$) following the application of a sequence of $V_{write}$ across a loop (as indicated by the spinning arrow). As in earlier experiments[41,47], the hysteretic switching is fully reproducible (two $G_{100}$ and two $G_0$ loops are superposed in Fig. 2c) and asymmetric. In particular, switching from ON to OFF requires a smaller $|V_{write}|$ than the opposite ($V_{write} \sim -1.5$ V vs. $V_{write} \sim 3$ V). The highest electroresistance $ER \equiv G_{ON}/G_{OFF}$ is observed for low $V_{read}$: $ER_{0mV} \sim 540$ while $ER_{100mV} \sim 100$. The temperature dependence of the conductance, shown in the inset of Fig. 2e for the ON and OFF states, also behaves differently at low and high $V_{read}$. At $V_{read} = 100$ mV (circles), a monotonous trend is observed across the entire temperature range: $G_{ON}$ is nearly constant and $G_{OFF}$ logarithmically decreases with decreasing temperature. However, for

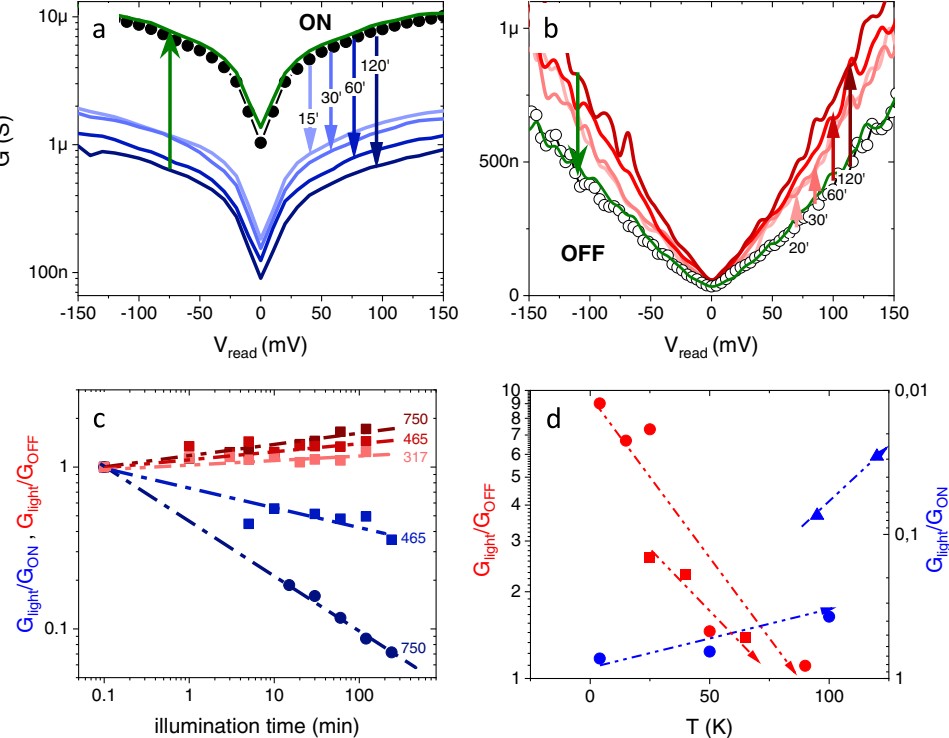

**Fig. 3 | Optical switching. a, b** Differential conductance as a function of $V_{read}$ before (black) and after illumination (colours) for increasing illumination times (see labels). In (**a**), the junction is set in the ON state prior to illumination, while in (**b**) the same sample is set in the OFF state prior to illumination. In both cases, visible light ($\lambda = 405$ nm) with optical power of $P = 750$ mW.cm$^{-2}$ was used. $T = 100$ K for the measurements in the ON state (**a**) and $T = 4$ K in the OFF state (**b**). The temperature effects are further discussed in (**d**). **c** Time dependence of the light-induced relative conductance variation $G_{Light}/G_{ON}$ and $G_{Light}/G_{OFF}$ in the ON (blueish curves) and OFF (reddish curves) state respectively, for several optical powers (different colour shades) as indicated by the labels (in mW.cm$^{-2}$). Different symbol shapes (circles/squares) correspond to different junctions. In all cases, visible light ($\lambda = 405$ nm) was used. $T = 100$ K for the measurements in the ON state and $T = 4$ K for those in the OFF state. **d** Temperature dependence of the relative light-induced conductance variation $G_{Light}/G_{ON}$ and $G_{Light}/G_{OFF}$ for different junctions (different symbols shapes) in the ON (blue) and OFF (red) state. The illumination conditions were: visible light ($\lambda = 405$ nm), with an optical power of $P = 750$ mW.cm$^{-2}$ in the ON state. In the OFF state, the samples were illuminated either with visible light and with an optical power of $P = 150$ mW.cm$^{-2}$ (red circles) or with UV light ($\lambda = 365$ nm) with an optical power of $P = 750$ mW.cm$^{-2}$ (red squares), for a fixed illumination time $t = 120$ min.

$V_{read} = 0$ mV (squares), a departure from those trends is observed below the superconducting critical temperature T$_C$ (which is lower in the OFF than in the ON state). As we demonstrated earlier[41], the greater *ER* and distinct temperature behaviour observed for $V_{read} = 0$ mV are explained by the opening of the superconducting gap, which reduces the electronic density of states at the Fermi level and dramatically decreases the tunnelling conductance (see supplementary Fig. S2). In summary, the YBCO/ITO junction presents similar tunnelling electro-resistance effects as those found earlier in YBCO/Mo$_{80}$Si$_{20}$ junctions, including a pronounced electroresistance enhancement in the superconducting state.

While at low temperatures both the ON and OFF states are stable over the duration of the performed experiments, for several hours, their behaviour differs at higher temperatures. The OFF state is indeed stable over time at any temperature, however, at sufficiently high ones the ON relaxes towards the OFF state. This is demonstrated in Fig. 2g, which shows the time-dependent normalized conductance switching $\triangle G_E(t)/G_{ON}(t=0)$ at different temperatures, with $\triangle G_E(t) = G(t) - G_{OFF}$. $\triangle G_E$ stays constant as a function of time at low temperatures (up to $\sim$ 80 K), but relaxes towards zero at higher ones, at a rate that increases as the temperature is increased. It does so by following a stretched exponential $\triangle G_E(t)/G_{ON}(t=0) = e^{-(t/\tau)^\beta}$, with $\beta \sim 0.21$ and $\tau$ a temperature-dependent time scale that shows Arrhenius behaviour with an activation energy of the order of 0.1 eV (see Supplementary Fig. S4 for further details). In summary, the observations imply that the OFF state is the system's ground state, while the ON state is metastable below $\sim 8$ K and relaxes towards the OFF at higher temperatures.

## Optical effects

The optical effects described below are the main finding of the present experiments and are divided into two classes. The first one corresponds to a persistent effect: a photoinduced switching (increase or decrease) of the junction's conductance level, analogous to the switching produced by $V_{write}$ pulses. The second one corresponds to a volatile effect: a photovoltage observed *during* illumination.

The photoinduced switching effects are summarized in Fig. 3. Figure 3a, b respectively display the differential conductance $G(V_{read})$ for the same junction after it has been set in the ON and OFF states and subsequently illuminated with visible light ($\lambda = 405$ nm) over a time ranging from 15 to 120 min (see labels). Strikingly, the photo-response is very different in each case. If the junction is set in the ON state before illumination (Fig. 3a), a gradual photo-induced decrease of conductance is observed as the illumination time increases (series of blue-shaded curves). Notice that for the longest illumination time (120'), the overall conductance drops by a factor of ten. Contrarily, if the junction is set in the OFF state before illumination (Fig. 3b), a gradual increase of the conductance is observed as the illumination time increases. Whatever the conductance level reached after illumination, the junctions can be reset to the ON/OFF state by the application of a positive/negative $V_{write}$ (see the green curves in Fig. 3a, b). It is important to notice that, while the electrical switching drives the junction all the way from ON to OFF and vice versa (Fig. 2f), optical switching drives the junction into intermediate states. As shown in Supplementary Figure S9, the intermediate states are virtually non-volatile at sufficiently low temperatures, but relax

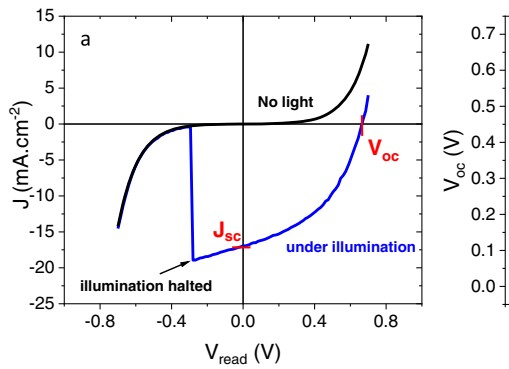
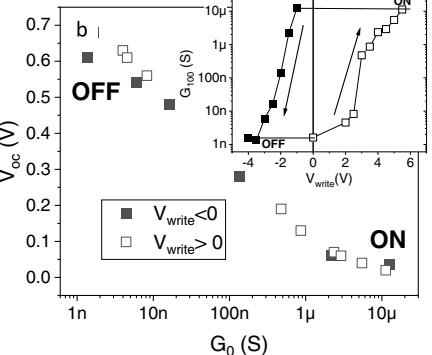

**Fig. 4 | Photovoltaic effect. a** Current density $J$ vs $V_{read}$ measured in dark and under illumination with visible light ($\lambda = 405$ nm), using an optical power of $P = 150$ mW.cm$^{-2}$ at $T = 4$ K for a junction previously set in the OFF state. The illumination is halted at the point indicated by the arrow. The open circuit voltage $V_{oc}$ and short circuit current $J_{sc}$ are highlighted. **b** Open circuit voltage $V_{oc}$ measured across the junction as a function of the remnant zero-bias conductance $G_0$ at $T = 4$ K as the junction is switched from the ON to the OFF state (solid symbols.) and the OFF to the ON state (hollow symbols) by application of $V_{write}$. The inset shows the corresponding $G_0$ as a function of $V_{write}$.

towards the OFF state at higher ones, at a rate that increases with temperature. This is as expected from the relaxation behaviour shown in Fig. 2g and Supplementary Fig. S4.

Figure 3c displays the relative change of the conductance in the ON and OFF state, $G_{light}/G_{ON}$ and $G_{light}/G_{OFF}$, as a function of the illumination time and for different optical powers (indicated by the labels), with $G_{light}$ (and $G_{ON}$ or $G_{OFF}$) the conductance after (before) illumination. Data are extracted for $V_{read} = 100$ mV from measurements as those in Fig. 3a, b. One can see that light effects are proportional to the illumination time and power (further illustration of the power effects is shown in Supplementary Fig. S11). That is, the device behaves as a photon integrator: the size of the photoinduced conductance variation increases as the number of photons shone on the device increases. Notice that the response is not linear (Fig. 3c is in double log scale).

In summary, whether illumination produces an increase or a decrease in the conductance level depends on the previous electrical history, and the size of the conductance variation depends on the number of photons shone on the device (optical history). The optical and electrical memristive responses are thus coupled, which results in new functionality in which light effects are bidirectional, cumulative and can be "erased" by application of a voltage pulse.

The photoinduced effects in the ON and OFF states also differ in their temperature dependence. This is shown in Fig. 3d, which displays the relative change of the conductance after illumination ($G_{light}/G_{ON}$ and $G_{light}/G_{OFF}$ at $V_{read} = 100$ mV) as a function of temperature, for various samples and illumination conditions (see figure caption). One can see that, in all cases, in the OFF state (red data points) the photo-induced changes are the largest (up to 1000%) at low temperatures and gradually decrease as the temperature is increased, until they become only a few per cent at around 100 K. Contrarily, in the ON state the effects are relatively small at low temperatures and gradually increase as the temperature is increased, reaching up to 5000% at ~ 100 K. Notice that there exists a range of intermediate temperature where both effects are similarly strong. The completely different temperature dependence suggests that optical switching arises in the ON and OFF states from different microscopic mechanisms.

Figure 4 illustrates the second class of photoinduced phenomena, namely the photovoltaic effects. Figure 4a shows the typical current $I$ vs $V_{read}$ measured in the OFF state, both in the dark (black curve) and under illumination with $\lambda = 405$ nm and $P \approx 150$ mW.cm$^{-2}$ (blue curve). As soon as the junction is illuminated, we observe a large photovoltaic effect, i.e. a shift downwards of the $I$-$V_{read}$ curve. This is caused by the appearance of a photocurrent that instantly disappears when the illumination is halted (see arrow). We define the short-circuit current $J_{sc}$ as the current measured when a zero bias is applied across the junction, and the open circuit voltage $V_{oc}$ as the voltage across the device when no electrical current is measured. While $V_{oc}$ (~ 0.5 V) is comparable to that observed earlier in cuprate based photovoltaic cells[48,49], $J_{sc}$ is several orders of magnitude higher[48,49] and indeed compares to that of silicon based devices[50]. Figure 4b displays $V_{oc}$ as a function of the junction's remnant conductance measured at $V_{read} = 100$ mV, $G_{100}$. This conductance level is set prior to illumination by application of $V_{write}$ pulses across the hysteretic switching loop (inset of Fig. 4b). The magnitude of $V_{oc}$ strongly depends on $G_{100}$, that is, on whether the sample is in the ON, OFF or intermediate states. In particular, one can see that the photovoltage is the highest at low conductance (OFF state), and logarithmically decays as the junction conductance is increased towards the ON state. Notice that $V_{oc}$ depends only on the conductance state $G_{100}$ and not on the electrical history, since the data points from the ON/OFF and OFF/ON switching branches (respectively solid and hollow symbols) collapse into a single master curve.

## Discussion
### Origin of electrical switching effects
The electrical switching behaviour observed in Fig. 2 is analogous to that of YBCO/Mo$_{80}$Si$_{20}$[41] and NdNiO$_3$/Mo$_{80}$Si$_{20}$[47] junctions, for which we demonstrated that the underlying mechanism is a voltage-driven oxygen exchange between the junctions' electrodes. Sketches of the interface in the ON and OFF state are shown in Fig. 2a. OFF is the ground state, in which YBCO is severely oxygen depleted at the interface due to the reduction of YBCO in favour of ITO oxidation, as it is demonstrated by the STEM and EELS analysis of the pristine YBCO/ITO interface (Fig. 2b−d). The oxygen-depleted YBCO layer is expectedly semiconducting with a 1.25 eV gap[51], and thus behaves as a tunnelling barrier under low voltage bias (a hundred meV). That layer's thickness is estimated at $5 \pm 1$ nm from the fits of the differential conductance to the BDR model for electron tunnelling (see supplementary Fig. S1). That is consistent with the STEM-EELS observations in Fig. 2b−d. The presence of this relatively thick oxygen-depleted YBCO yields the lowest conductance state (hollow symbols in Fig. 2e). The application of a sufficiently high $V_{write} > 0$, comparable to the different between electrochemical potentials $\triangle E_0 = E_0(Cu) - E_0(In) \approx 2.9$ eV, reverses the redox reaction, driving oxygen back into the interfacial YBCO and hence thinning down the oxygen-depleted YBCO layer[41]. This increases the conductance by orders of magnitude, leading to the ON state (solid symbols curve in Fig. 2e). The system can be driven again into the OFF state by the application of a negative voltage. Notice

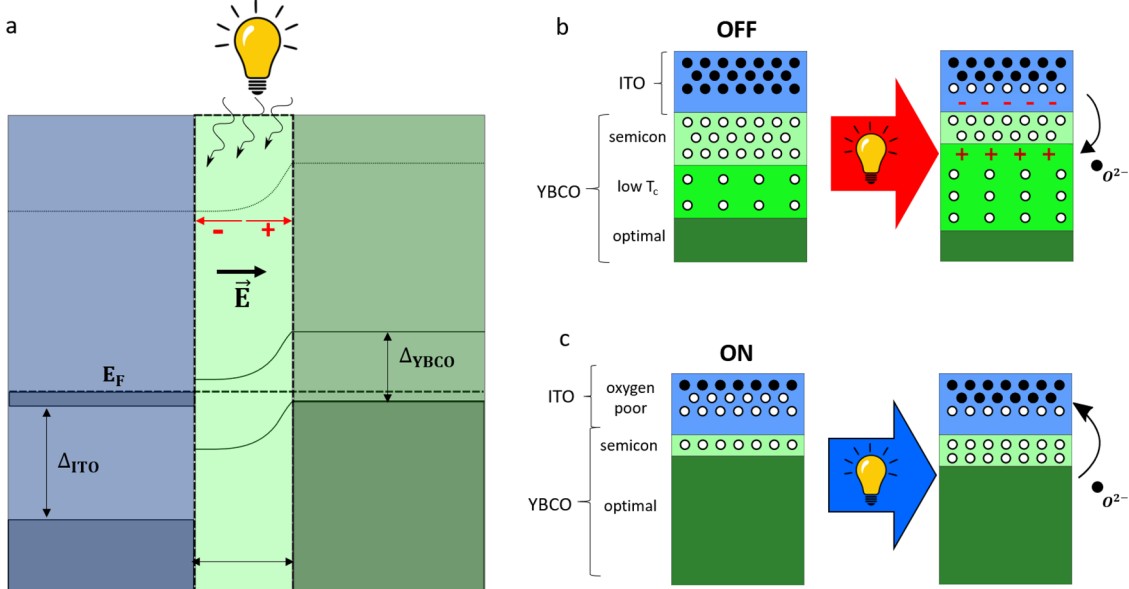

**Fig. 5 | Microscopic Model. a** Energy band diagram of the YBCO/ITO interface where ITO (blue) is a degenerate n-type semiconductor and oxygen-depleted YBCO (green) is a p-type semiconductor. A space charge layer, characterized by a built-in electric field at the interface, is indicated by the light green region. $\triangle_{ITO}$ and $\triangle_{YBCO}$ represent the electronic band gaps of ITO and YBCO respectively. **b, c** Schematic representation of the ITO/YBCO interface during optical switching. The full and hollow circles indicate oxygen atoms and vacancies respectively. In the OFF state, see **b**, the interfacial YBCO is highly oxygen deficient. Conversely, the ITO oxygen content is optimal. Upon illumination, the photovoltage $V_{OC}$ leads to an accumulation of holes (red crosses) in the interfacial YBCO, which promotes local oxidation by migration of $O^{2-}$ atoms. When the junction is set in the ON state, see **c**, there is no photovoltage. Oxygen has been driven from the ITO into the YBCO, leaving an interfacial, oxygen-depleted ITO. This is an out-of-equilibrium state because ITO has a lower reduction potential than YBCO. Thus, oxygen tends to migrate into ITO, leading to a natural relaxation from the ON towards the OFF state. This is accelerated by illumination.

that the magnitude of the voltage required to trigger the ON→OFF switching is lower than for the OFF→ON one, since for the former the underlying reaction is spontaneous from an electrochemical standpoint. Thus, the finite $V_{write}<0$ is required only to accelerate oxygen ion diffusion. This is consistent with the spontaneous ON →OFF relaxation observed in the presence of thermal activation (Fig. 2g and supplementary Fig. S4). It is worth noting that the activation energy extracted from the relaxation analysis $E_a \sim 0.1$ eV is well in the range of the values found in oxygen exchange reactions in YBCO[52], which further supports the redox scenario. Another piece of evidence supporting that the large conductance switching is produced by oxygen exchange between YBCO and ITO is that it is suppressed when ITO is replaced by a material which does not oxidize (see supplementary Fig. S3).

## Origin of the photovoltaic effect
The key to understanding this effect, sketched in Fig. 5, is that the strongly oxygen-depleted YBCO is a p-type semiconductor with a $\sim 1.25$ eV gap[51], and the adjacent ITO is a degenerate n-type semiconductor with a $\sim 3.5$ eV gap[53]. Thus, the junction interface can be seen as a p-n junction. This is supported by the diode behaviour observed in the I(V) of the junctions under high voltage bias (see supplementary Fig. S10). In consequence, a photovoltaic effect is naturally expected. The different work functions $\phi_{ITO} = 4.5$ eV[54] and $\phi_{YBCO} \approx 5.5 - 6$ eV[55] leads to a space charge layer (SCL) with a built-in electric field pointing towards the YBCO, as represented Fig. 5a. In this scenario, upon illumination, electrons and holes photogenerated in the SCL respectively flow towards the YBCO and the ITO, leading to the observed photocurrent (photovoltage in open-circuit configuration). We can obtain a rough estimate of the SCL thickness using[56]

$$W = \sqrt{(N_A + N_D/N_A N_D) 2\varepsilon_s V_{bi}\backslash e}$$ where $N_A \sim 8\,10^{19} \text{cm}^{-3}$ and $N_D \sim 5\,10^{20} \text{cm}^{-3}$ are respectively the carrier densities in oxygen-depleted YBCO[57] and in ITO[58], and an upper limit for built-in voltage

$V_{bi}<1.5$ V can be inferred from the difference between work functions. From this, $W<5$ nm which is consistent with the fact that the conduction is dominated by electron tunnelling in the low-bias regime, as well as with the tunnel barrier thickness $\sim 5$ nm obtained from the fitting to the BDR[45] model (supplementary Fig. S1). The scaling of $V_{oc}$ with the logarithm of the junction's conductance state $G_{100}$ shown in Fig. 4b is also consistent with the discussed scenario. Indeed, if one considers that the current across the junction and the conductance are roughly proportional in the low $V_{read}$ regime, the behaviour of Fig. 4d is as expected from the Shockley relation[59] $V_{oc} = \frac{k_b T}{\eta q} \ln(\frac{I_{sc}}{I_0} - 1)$ where $\eta$ is the ideality factor, $q$ is the electric charge, $I_{sc}$ is the photocurrent and $I_0$ is the reverse saturation current in the dark. This explains why $V_{oc}$ is the largest in the low conductance (OFF) state, in which $I_0$ is the lowest, as well as the logarithmic decay of $V_{oc}$ as the conductance (and $I_0$) increases. Notice finally (Supplementary Fig. S8) that a finite $V_{oc}$ appears for wavelengths in the range 365–400 nm (3.4–3 eV), i.e. for photon energies above the YBCO charge transfer gap. This is as expected since metal-ligand charge transfer excitations into $CuO_2$ plane states at energy losses >3 eV have been reported in oxygen-depleted YBCO[60].

## Origin of optical switching effects
After the junction is set in the ON state by the application of a positive $V_{write}$, illumination leads to a strong conductance decrease (Fig. 3a) which is persistent at low temperatures. To understand this behaviour, we start by recalling that the ON state (sketch in Fig. 2a) is out of equilibrium from the electrochemical point of view, since the material with the lowest reduction potential (ITO, see Supplementary Table 1) presents a strong oxygen deficiency near the interface while most of top YBCO layers are fully oxygenated. This explains why the ON state naturally relaxes towards a lower conductance one, as demonstrated in Fig. 2g and Supplementary Fig. S4. Microscopically, relaxation occurs as oxygen migrates back

into ITO, driving the system towards its ground state, as shown Fig. 5c –with the interfacial YBCO oxygen-depleted in favour of ITO oxidation (see STEM in Fig. 2b, c). However, this requires overcoming the barrier for ion diffusion. While this is thermally activated in the dark (see Supplementary Fig. S4), at low temperatures (below ~ 90 K) oxygen mobility is low and the relaxation is very slow, making the ON state virtually non-volatile (Fig. 2g). Upon illumination, the relaxation is dramatically accelerated: as demonstrated by Supplementary Figure S5, illumination makes the system relax as it would do ~ 200 K above the actual temperature. That is, illumination activates ion diffusion, expectedly due to phonon excitation by light absorption. This is consistent with the fact that the photoinduced conductance decrease does not show a marked wavelength dependence in the visible range (Supplementary Figure S6), and that IR light produces strong effects (Supplementary Figure S7). As oxygen migrates from YBCO into ITO thanks to optical activation (see the sketch in Fig. 5c), the severely oxygen-deficient YBCO layer at the interface thickens, leading to the decrease of the tunnelling conductance (Fig. 3a)—which, at low T, is stopped as soon as illumination is halted. In summary, the photoinduced conductance decrease in the ON state reflects light-activated oxygen mobility. It is worth noting that this mechanism was proposed earlier in the literature of cuprates[61,62] to explain a different type of persistent photoconductivity in thin films.

Once the junction is set in the OFF state by a negative $V_{write}$, the interfacial YBCO is severely oxygen depleted and ITO is oxygen-rich (sketch in Fig. 2a). In this state, and contrary to the ON state, a conductance enhancement is observed after illumination (Fig. 3b). That is explained by a key specificity of the OFF state: the large positive photovoltage $V_{OC}$ measured *during* illumination (see Fig. 4). Indeed, a causal relationship exists between $V_{OC}$ during illumination and the conductance enhancement observed after illumination. This is documented in the supplementary Figs. S7 and S8, which demonstrate that the conductance enhancement is observed only in the presence of a photovoltage $V_{OC}$, and scales with its size. As discussed above and sketched in Fig. 5a, the photovoltage reflects a hole accumulation at the YBCO side of the space charge layer. This accumulation changes the doping of the interfacial YBCO, locally enhancing the number of holes per copper atom and thus taking the oxidation state $Cu^{+2} + h^+ \rightarrow Cu^{+3}$ beyond the level corresponding to the actual oxygen content. This favours the re-oxygenation of the interfacial YBCO by migration of $O^{2-}$ ions (sketch in Fig. 5b). This mechanism is analogous to the well-known photocatalytic effect produced as photocarriers promote redox reactions in the vicinity of pn-junctions[63,64]. $O^{2-}$ ions can migrate from the oxygen-rich interfacial ITO, where the photo-induced electron accumulation (Fig. 5a) allows $O^{2-}$ to be released without changing the In oxidation state ($In^{+3} + e^- \rightarrow In^{+2}$). Notice that, consistently, the $O^{2-}$ migration from ITO is also favoured by the $V_{OC}$ polarity, which is the same required to trigger that migration by application of $V_{write}$ pulses. In addition, we cannot discard that some $O^{2-}$ ions diffuse into the interfacial YBCO from deeper YBCO layers that present higher oxygen content, given the sharp gradient of oxygen content expected from $V_{write} < 0$. In summary, the positive photovoltage drives oxygen ions towards the oxygen-depleted interfacial YBCO, similarly as an applied $V_{write} > 0$ does. Re-oxygenation leads to a thinning of this oxygen-depleted YBCO layer, which is the electron tunnelling barrier, thus enhancing the electrical conductance (Fig. 3b).

To summarize, light promotes oxygen migration, either in or out of the interfacial YBCO depending on whether the junction is previously set in the ON or the OFF state by $V_{write}$ pulses. Oxygen migration is favoured by optical activation over the barrier for ion diffusion and driven either by (i) (ON state) the redox reaction dictated by the higher reduction potential of YBCO when it is oxygen-rich and ITO is oxygen deficient; or (ii) (OFF state) by the positive photovoltage

and ensuing hole/electron accumulation at both sides of the interface, which reverses the redox reaction. While (i) and (ii) are obviously antagonistic, (ii) is not present in the ON state and becomes relevant gradually as the junction is driven into the OFF state and the photovoltage $V_{OC}$ increases (Fig. 4b). This explains why light cannot drive the junction all the way from the ON into the OFF state (nor vice versa), but only into an intermediate level that depends on the predominance of either (i) or (ii), their balance being controlled by temperature (Fig. 4d) and/or the optical wavelength (supplementary Figs. S6, S7 and S8).

Aside from their fundamental interest, the described effects are also technologically relevant for various reasons. In addition to the unique photo-memristive behaviour, the devices studied here –made of a single interface– are of ultimate simplicity when compared to approaches that involve more complex geometries[17,20] and photo-active materials[17,23]. It is worth noting that photo-memristive effects similar to the ones reported here for superconducting YBCO should be observed if this is replaced by other conducting oxides, provided that the junction is formed by materials having a different reduction potential (to promote oxygen exchange) and that a p-n junction is formed at the interface (allowing for photovoltages). This is the case, for example, of junctions based on $NdNiO_3$, for which we have recently reported[47] electrical switching effects analogous to those demonstrated here. The fact that the hereby demonstrated photo-memristive effects are based on a high-temperature superconductor enhances their technological potential. First, while the switching effects are observed at any temperature, they become much stronger below the superconducting $T_C$. This, together with the vanishing Joule dissipation, expectedly facilitate miniaturization and the layout of dense memristor arrays with high interconnectivity, as required for neuromorphic computing[13,15]—an emergent area also within superconducvitiy[65,66]. Furthermore, the photo-memristors realized here could be naturally implemented in conventional superconducting electronics[32], a thriving field, particularly after the advent of Josephson devices based on high-temperature cuprates[33–35] as the one used here, which allow operation above liquid nitrogen temperatures. One could for instance cite applications such as quantum antennas[36] and logic circuits[37], in which non-volatile memory is a grail[67] and photo-sensitivity was up to now unavailable. The incorporation of these functions should refashion and greatly broaden the range of applications of such Josephson circuits.

## Methods

### Sample fabrication

The 30 nm-thick c- axis $YBa_2Cu_3O_{7-\delta}$ (YBCO) films were grown by pulsed laser deposition (PLD) on $SrTiO_3(001)$ (STO) substrates, at 695° C, in an $O_2$ atmosphere of 0.36 mbar, using a 238 nm KrF excimer laser with an energy density ~1 J.cm$^{-2}$ and ~ 5Hz repetition rate. After deposition, the films were cooled down in a pure oxygen atmosphere (800 mbar) to obtain optimal oxygen stoichiometry. Standard photolithography was used to fabricate studied junctions. First, a layer of photoresist was spin-coated onto YBCO films in which square openings (10–400 $\mu m^2$) were patterned to define the contact area between YBCO and ITO. This first layer of photoresist was hard-baked to make it immune to further illumination and the solvents (developers, acetone, isopropanol) used in the subsequent processing, and thus persistent. A second layer of photoresist was then spin-coated, on which larger (200 × 600 $\mu m$) openings were defined to create electrical contact pads aligned with the small square openings in the first layer. Then the 100 nm thick ITO film was deposited by RF sputtering at room temperature, in an atmosphere mixture of argon and oxygen gas (3 sccm of Ar 97%/$O_2$3% premix plus 43 sccm of pure Ar), at a deposition pressure of $6x10^{-3}$ mbar and using a commercial target of $In_2O_3$ 90%/$SnO_2$10%. This was followed by a lift-off of the 2$^{nd}$ photoresist layer. Two wires were attached to the YBCO film (junction's

bottom electrode) and one to the top ITO contact pad (top electrode) using Aluminium wire and a wedge bonding machine. This allows for performing three-probe electrical measurements of the YBCO/ITO junctions.

## Transport measurements

The writing voltage was applied across the junctions by ramping it from 0 V up to $V_{write}$ then back to 0 V. Current vs bias voltage $V_{read}$ characteristics in the remnant state were subsequently measured using a Keithley 2450 sourcemeter, with the bias across the junction $|V_{read}| \leq 200$ mV measured using a Keithley 2182 Nanovoltmeter. The differential conductance ($V_{read}$) was obtained by numerically differentiating the measured current-voltage characteristics. In all measurements, the top ITO electrode is grounded.

## Scanning Transmission Electron Microscopy (STEM)

For STEM characterization, YBCO-ITO bilayers were characterized using an aberration-corrected JEOL JEM-ARM 200cF electron microscope operated at 200 kV and equipped with a cold field emission gun and a GATAN Quantum electron energy loss spectrometer. For spectral imaging, the electron beam was scanned along the region of interest and an EEL spectrum was acquired for every pixel.

## Data availability

The data that support the findings of this study are available from the corresponding author upon reasonable request.

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

## Acknowledgements

Work supported by ERC grant N° 647100 "SUSPINTRONICS", ERC grant N° 966735 "SUPERMEM", French ANR grant ANR-17-CE30-0018-04 "OPTOFLUXONICS", COST Action CA 21144 superqumap, and Spanish AEI PID2020-118078RB-I00. J.S. thanks the D'Alembert program funded by the IDEX Paris-Saclay, ANR-11-IDEX-0003-02, for financing a stay at Unité Mixte CNRS/Thales. We (J.S., J.E.V.) acknowledge funding from Flag ERA ERA-NET To2Dox project. J.S. acknowledges AEI through grant PID2020-118078RB-I00. G.S.-S. acknowledges financial support from Spanish MCI Grant Nos. RTI2018-099054-J-I00 (MCI/AEI/FEDER, UE) and IJC2018-038164-I. Electron microscopy observations were carried out at the Centro Nacional de Microscopia Electronica, CNME-UCM.

## Author contributions

The experiments were conceived by J.E.V. with inputs from R.E.H., A.S. and J.S. The materials were grown by A.S., J.B., S.J.C., and J.C.. The junctions were lithographed by R.E.H. and V.H. with the support of S.M. The transport measurements and data analysis were carried out by R.E.H., V.H. and V.R. K.S. contributed to the analysis of the data. The microscopy studies were carried out by G.S.-S. All the named authors, A.L. and J.T. discussed the experimental results. The microscopic model of the photo-response was put together by R.E.H., J.S. and J.E.V. The paper was written by R.E.H. and J.E.V. with inputs from all other authors. The overall project was supervised by J.E.V.

## Competing interests

The authors declare no competing interests.
