## [Peer Review File · Nature Communications]

Reviewers' comments:

Reviewer #1 (Remarks to the Author):

The authors reported a resistive switching effect at a superconductor/semiconductor junction, where the conduction on-off ratio can be modulated by both electric field and photo-illumination. They claim that the microscopic mechanism at play is a reversible nanoscale redox reaction between both materials, whose oxygen content determines the electron tunnelling rate across their interface. Oxygen exchange is controlled here via illumination by exploiting a competition between electrochemistry, photovoltaic effects and photo-assisted ion migration. Although the data quality is high, the mechanism behind the phenomena is lack strong evidence. Additionally, the writing is poor, especially they garble several concepts. In short, the paper do not have significant results to be published at Nature Communications. The detailed comments are as follows:

1. The description of “superconducting” in the title is misleading. Maybe, they can change to “photo-memristor based on a superconductor/semiconductor junction”?
2. The author claim “a tunneling photo-memristor” in the abstract. This is not demonstrated by any data. Actually, the observation is a resistance switching effect at a p-n junction. The conduction might be Thermionic emission or Poole-Frenkel emission.
3. How to demonstrate the oxygen exchange between the materials?
4. The photo-illumination can induce photovoltaic effect at the junction, but how can it induce electrochemistry and ion migration? How to know there occurs electrochemistry and ion migration? And why these phenomena can induce oxygen exchange?
5. What significant role does the superconductor play? What if the superconductor YBCO is replaced by other conducting oxides, such as (La,Sr)MnO₃ or SrRuO₃? A detailed discussion might be required.

Other minor points:

1. The reference format is not consistent.
2. The descriptions in many places are too long and should be simplified, especially for the figure caption.

Reviewer #2 (Remarks to the Author):

In general the photomemristive effect with opposite effects in the ON and OFF states is something novel, and a good finding. The interplay between atomic diffusion, redox reactions, photons and electron transport is cute. However, I am not very convinced of the technological merits that the authors are compelled to mention. For any sort of merits in a neuromorphic architecture, one needs to be able to use these devices at room temperature, or else there is a lot of overhead required to cool these devices (at a circuit level). That will kill any potential benefits.

The authors may instead argue for architecture in photon based quantum computing that would still require electrooptic effects, and low temperatures are OK there.

Having said that to make a decision on the scientific merits of the paper I need some clarifications from the authors, and I invite them to answer the following :

a. I am not quite sure what is the use of YBCO here. Can this photomemristive phenomena not work with any oxide that undergoes topotactic reactions (LSMO, for e.g.).

b. Is the advantage of a superconductor only that ON-OFF ratios enhance below T_c ? I couldn't clearly understand the importance of T_c in photoinduced effects.

c. About figure 2d which shows the relaxation of the ON state, does it relax finally to an OFF state? Or does it asymptotically settle down at a value of $\Delta G_e(T)/\Delta G_{on}$, which is non zero. Even at 250 C, I don't see this approaching 0. Which means that ON state is kinetically frozen for a long time at room temperature. That's OK from a device perspective.

d. When ON state sees photons its resistance increases (non-volatile fashion). I am guessing once light is removed, this should also decay with time (just like it happens in the ON state under dark conditions). Please report that data, and comment on the retention. Same applies for OFF state. If the decrease in resistance is due to a photovoltaic effect, which is volatile, the so-called "non-volatile" increase in resistance should have a time scale associated with it, before which it recovers back to the starting OFF state. Please present retention data on both OFF and ON states after illumination is removed (at different read voltages).

e. The idea of a pn junction to explain volatile PV effect is very interesting. However, the I-V transport shown in figure 4a doesn't seem to fit with the behavior of a diode. Even if the electron transport were more complicated than a simple diode, I expect some sort of asymmetry between negative and positive voltages owing to the presence of a pn junction. Rather these I-V curves look

like SCLC type conduction (to speculate). Please comment, and prove that there is a pn junction through I-V measurements, and if necessary simple circuit modeling.

f. For the ON state resistance to decay with photons, what exact redox reactions are important for this. Please detail the chemistry. And does any wavelength of light work? Or should one be in a wavelength where carriers are generated (UV)? In other words, infrared radiation may not do anything (if heating is excluded).

g. Furthermore, the suggestion that photons somehow enhance diffusion (without heating), would mean momentum transfer from photons to oxygen vacancies. Photon momentum is negligible, that photomigration is a difficult process to imagine. If this is not the case, what do photons do to enhance the ion migration?

h. Since the ON state increases in resistance with illumination, OFF state decreases with illumination, if shone long enough on the ON state say, the increase in resistance should slowly subside and there should an upturn after which resistance decreases. Do the authors see this? What are the typical time scales? If that is not the case then ON state is not really relaxing to an OFF state, but rather relaxing to a different state.

g. For photon integration or photon counting, one needs to show how the detected parameter (ΔG) changes with the photon intensity. If this is linear over a large range, it is a good detector. Here we don't know what the characteristics of ΔG vs photon power are. If the authors are to claim that this will be good photon counter, they must first show this relation. ΔG may saturate at larger powers, and that may not be that useful.

Reviewer #3 (Remarks to the Author):

This manuscript deals with an interesting concept for memristive devices. The authors show that the interface between a bottom YBa₂Cu₃O₇ (YBCO) electrode and an indium-tin-oxide (ITO) top electrode can behave as a tunnelling photo-memristor. The authors argue that the physical properties of the oxide heterointerfaces are determined by electronic and/or ionic reconstructions.

More interesting, both electrical and optical stimuli are shown to be able to alter the resistance states, depending on both optical and/or electrical history. Such light sensitivity to electrical

memristors might open possibilities for complex computational applications involving both electrical and optical switching in a single passive component.

Although the device shows clear bimodal memristive behavior, and the concept might be considered novel, I feel that the manuscript lacks detailed understanding of the behavior. The authors assume that the physical properties of the oxide heterointerfaces are determined by electronic and/or ionic reconstructions. Mainly hand-waiving arguments have been described in the manuscript. In general, both the measured effects and the suggested mechanisms are described sufficiently. However, the authors did not provide evidence for these mechanisms. Furthermore, the crystalline quality and the interface characteristics (such as roughness) are very much dependent on the synthesis routes and reproduction of these devices might be hard in other labs. More info on materials characteristics of the interface might be necessary. To warrant publication in Nature communications, I feel that the manuscript needs to be significantly improved.

Interfaces in heterostructure with YBCO (in contact with for instance metals or other oxides) have been studied a lot in the last decades, and redox mechanisms have been shown to be responsible for altering the properties of the interface. Several other experimental results, for instance from spectroscopic and transport measurements, indicate the role of redox driven reactions. As a general feature for cuprate superconductors, the Cu-O bond is weak, and redox reactions easily occur at the cuprate interface with the contact layer. Such reactions can result in electron doping, oxygen (vacancy) migration, complete destruction of superconductivity in the interface region, or modification of the YBCO microstructure.

Next, electric field manipulation have been shown as an effective method to tune the redox-induced effects, and these experiments show that electrochemical modification of interfaces results in a nontrivial spatial profile of the oxygen vacancy distribution close to the interface.

Furthermore, optical modulation of the resistance of memristors is not new and photomemristors and their behavior, such as MoS₂ nanosphere-based and Al/perovskite FAPbBr₃/ITO heterostructures, have been demonstrated before. Also in such structures, modification of the resistance state is due to ionic and/or electronic effects.

In conclusion, I feel that the authors show an interesting device, and the optical/electronic resistive switching behavior is well-described. The manuscript however lacks materials science aspects of the device and sufficient proof for the suggested mechanisms. I therefore propose to reject this manuscript for publication in nature communications.

Response to Referee #1

We thank the Referee for reviewing our manuscript. We are glad that he/she acknowledges the high quality of the reported data. Notwithstanding, he/she thinks that there is not enough evidence for the “mechanisms behind the phenomena”, and suggests that the writing should be improved. To address the Referee’s criticism, and following her/his suggestions, we have conducted several complementary experiments and substantially revised the manuscript. A point-by-point response to the referee’s report, together with a list of changes, can be found below. We think that this effort has allowed us to significantly improve the paper. We hope that the referee finds the revised version suitable for publication in *Nature Communications*.

*“The authors reported a resistive switching effect at a superconductor/semiconductor junction, where the conduction on-off ratio can be modulated by both electric field and photo-illumination. They claim that the microscopic mechanism at play is a reversible nanoscale redox reaction between both materials, whose oxygen content determines the electron tunnelling rate across their interface. Oxygen exchange is controlled here via illumination by exploiting a competition between electrochemistry, photovoltaic effects and photo-assisted ion migration. **Although the data quality is high, the mechanism behind the phenomena is lack strong evidence.** Additionally, the writing is poor, especially they garble several concepts. In short, the paper do not have significant results to be published at *Nature Communications*. The detailed comments are as follows:*

Referee’s comment # 1 : *“1. The description of “superconducting” in the title is misleading. Maybe, they can change to “photo-memristor based on a superconductor/semiconductor junction”?*

Our response: We thank the referee for this suggestion.

Changes made: The title has been changed to “Bimodal ionic photo-memristor based on a high-temperature oxide superconductor/semiconductor junction”

Referee’s comment # 2: *“2. The author claim “a tunneling photo-memristor” in the abstract. This is not demonstrated by any data. Actually, the observation is a resistance switching effect at a p-n junction. The conduction might be Thermionic emission or Poole-Frenkel emission”.*

Our response: We regret that we have not been clear enough in detailing the basis for that claim. The referee is right in that we observe resistance switching at a p-n junction. However, the space charge layer (SCL) is very thin (in particular, ~5 nm thick according to the estimates given in the manuscript). In these conditions, one expects that the low-bias conductance is dominated by electron tunneling. This expectation is supported by various experimental facts.

In particular, the electrical transport behavior observed in Fig. 2 of the manuscript is similar to that of YBCO/MoSi junctions and reported in our earlier paper Nat. Commun. 11, 658 (2020), in which we showed that tunneling is the governing conduction mechanism based on two key hallmarks:

- 1) The fact that in the low-bias regime ($V_{\text{read}} \lesssim 150$ mV), the conductance vs bias voltage data in the OFF state can be fitted using the Brinkman-Dynes-Rowell model with physically reasonable parameters (barrier thickness and height). Notably, the barrier thickness deduced from the tunneling conductance fits is consistent with the estimated SCL thickness, and with the thickness of the severely depleted YBCO layer measured with the new STEM experiments.

- 2) The presence of a zero-bias conductance dip that appears across the superconducting transition temperature and deepens/widens as the temperature is further decreased. This dip reflects the opening of the superconducting gap in the DOS of the YBCO electrode [see our previous paper Nat. Commun. 11, 658 (2020)], and its observation is considered definitive proof of electron tunneling [see e.g. Appl. Phys. Lett. 77, 1870 (2000)].

Both (1) and (2), are present in the samples discussed in the manuscript. We have emphasized this by adding supplemental material in the revised version (see details below).

Furthermore, following the referee's suggestion, we have ruled out the possibility that the conduction might be dominated by either thermionic or Pole-Frenkel emission. Indeed, neither of those mechanisms can explain the conductance data.

In the Poole-Frenkel case, one expects that the conductance $\frac{I}{V}$ fulfills $\ln\left(\frac{I}{V}\right) \propto \frac{e}{k_b T} \left(2\sqrt{eV/4\pi\epsilon d} - \phi_B\right)$, with ϵ the permittivity, T the temperature, ϕ_B the conduction barrier in the absence of bias V , and d the barrier thickness. Thus, $\ln(I/V) \sim A + B\sqrt{V}$, A and B being bias independent and directly proportional to the inverse temperature (see e.g. Sze, *Physics of Semiconductor Devices*, 2nd edition). We display in the figure below typical data for our junctions. This figure shows, on the one hand (main panel), that the data does not show Poole-Frenkel behavior, since at low bias there is a departure from the expected linear relation (dashed line). On the other hand, as seen in the inset, the slope B within the bias range in which a linear relationship is observed does not show the expected temperature dependence, since B is not directly proportional to $1/T$. Thus, the experimental data is not well explained by Poole-Frenkel emission.

A similar analysis for the thermionic emission case is shown below, based on the same experimental data. Here one would expect $\ln(I) \sim A + B\sqrt{V}$, with B being proportional to the inverse temperature [see e.g. Appl. Phys. Lett. 117, 222104 (2020)]. In this case, the

disagreement between the expected behavior and the experimental data trend is even stronger, as the linear relation is not observed for any bias range:

In summary, we conclude that electron tunneling is the dominant conduction mechanism based on:

- a) the fact that the I - V characteristics in the low-bias regime of interest behave as expected for electron tunneling, but not as expected for Poole-Frenkel nor thermionic emission;
- b) the observation of the superconducting gap, expected for electron tunneling into a superconducting electrode.
- c) the consistency of the parameters extracted from the conductance tunneling analysis and the microscopy experiments.

We would like to end up stressing that, while identifying the conduction mechanism across the interface is important for the sake of rigor, whether it is dominated by tunneling or other mechanisms is not relevant to the novel behavior we report in this paper (photo-memristive response) nor to the physical processes behind the photo-response.

Changes made:

- 1) We have included a new supplementary Figure S1 that shows typical fits of the conductance vs. bias to the BDR model, which can be made in the normal state of YBCO. These fits allow for estimating the tunnel junctions' parameters (barrier thickness and height) that are reported in the main text.

- 2) We have included a new supplementary Figure S2 that displays a set of conductance vs. bias curves at different temperatures. As detailed in the figure caption that set of data behaves as expected for electron tunneling into a superconductor, which supports electron tunneling as the governing tunneling mechanism in the studied junctions.
- 3) The above arguments are included in a new paragraph at the bottom of page 4 (marked in yellow).

Referee's comment # 3: "How to demonstrate the oxygen exchange between the materials?"

Our response: We regret this was not clear in the manuscript. We have extensively documented in two previous publications, the one already cited in the manuscript [Nat. Commun. 11, 658 (2020)] and the second one recently published [Adv. Sci. 2201753, 1 (2022)], that oxygen exchange produces in YBCO/Mo₈₀Si₂₀ and NdNiO₃/Mo₈₀Si₂₀ junctions the same electrical switching behavior as in the YBCO/ITO junctions studied here (the key novel finding reported in the present manuscript is photo-memristive behavior). Because the electrical switching is otherwise identical, and all those junctions have the same structure and share the same key ingredients (an interface between a complex oxide and a second material with a lower reduction potential), one can hardly imagine that the underlying switching mechanism may be different in the present experiments. The only difference between the junctions studied in [Nat. Commun. 11, 658 (2020)] and those in the present experiments is that we have replaced Mo₈₀Si₂₀ (a superconductor) with ITO (a transparent semiconductor).

ITO owes its conducting properties to electron doping by oxygen vacancies. It is thus a natural oxygen vacancy reservoir. On the other hand, in YBCO, the large reduction potential of Cu atoms in the CuO chains is responsible for its known tendency to be reduced which is further favored by large oxygen mobility (small barrier for long-range diffusion). A junction between both materials thus spontaneously favors oxygen diffusion from YBCO into ITO, a process that may be reverted by an external electric field.

Furthermore, (as also pointed out by Referee 3) YBCO is known for its redox activity due to oxygen exchange reactions. Redox kinetics of YBCO is dominated on the one side by the relatively high oxygen diffusivity in the bulk (particularly in the ab plane) [see refs A1- A4 below] and by the efficient oxygen surface exchange reactions [see ref. A5 below]. Interestingly the low activation energies of characteristic times of oxygen exchange reactions [A5] are similar to the ones obtained in this work (new Supplementary Figure S4), which further supports that oxygen transport is the dominant mechanism of the effects discussed here. To further strengthen the case, in the revised version we show additional experiments that prove oxygen exchange across the YBCO/ITO interface and also that, to obtain large resistance switching effects, one has to interface YBCO with a material avid of oxygen. In particular,

- a) We now include microscopy (STEM) and spectroscopy (EELS) of the YBCO/ITO interface (new Fig. 2b), which demonstrates a change of the oxidation state of Cu in YBCO within a few nanometers from the interface, supporting oxygen migration into ITO as well as the scenario sketched in Fig. 2a to explain the resistive switching effects.
- b) We have included in the Supplementary Information experiments in which ITO is replaced by Au, which does not oxidize, thus precluding the observation of large resistive switching effects.

[A1] L. Chen, C.L. Chen, A.J. Jacobson, Electrical conductivity relaxation studies of oxygen transport in epitaxial YBa₂Cu₃O_{7- δ} thin films, IEEE Trans. Appl. Supercond. 13 2882–2885 (2003)

[A2] J. Maier, G. Pfundtner, Defect chemistry of the high-T_c-superconductors, Adv. Mater. 3 292–297 (1991)

[A3] K. Yamamoto, B.M. Lairson, J.C. Bravman, T.H. Geballe, Oxidation kinetics of YBa₂Cu₃O_{7- δ} thin films in the presence of atomic oxygen and molecular oxygen by in-situ resistivity measurements, J. Appl. Phys. 69 7189–7201 (1991)

[A4] A. Gupta, Oxidation kinetics of YBa₂Cu₃O_{7- δ} thin films in different oxidizing ambients, Appl. Surf. Sci. 64 103–110 (1993)

[A5] P. Cayado, C.F. Sanchez-Valdes, A. Stangl, M. Coll, P. Roura, A. Palau, T. Puig, X. Obradors, Untangling surface oxygen exchange effects in YBa₂Cu₃O_{6+x} thin films by electrical conductivity relaxation, Phys. Chem. Chem. Phys. 19 14129–14140 (2017)

Changes made:

- 1) We have included new Figs. 2b, 2c & 2d which demonstrate the reduction of YBCO near the interface, as well as a new paragraph discussing the microscopy data supporting oxygen exchange between two materials and how that is explained by electrochemical arguments (see marked text in the 1st paragraph of page 4).
- 2) We have included a new supplementary Fig. S3 that shows that the large resistance switching effects are obtained only if YBCO is interfaced with a material that has a higher tendency to oxidize.
- 3) We have rewritten the text to accommodate the information given by 1) and 2), as well as to point at earlier literature that extensively discusses how oxygen exchange produces the electrical switching effects observed in ITO/YBCO junctions studied here. The rewritten text is found in the discussion section, between pages 7 and 8 (marked in yellow).

Referee's comment # 4: *“The photo-illumination can induce photovoltaic effect at the junction, but how can it induce electrochemistry and ion migration? How to know there occurs electrochemistry and ion migration? And why these phenomena can induce oxygen exchange?”*

Our response: We thank the referee for pointing out that this central question, one of the key novelties of the paper and largely discussed in the manuscript, remained unclear. As we detail below, in the revised version we have rewritten and extended the discussion –as well as included new experiments– to better explain the different photoinduced effects.

Our experiments show that illumination produces a bidirectional nonvolatile switching between conductance states, similar to that produced by applying electric pulses. As discussed above and as we demonstrated in Nat. Commun. **11**, 658 (2020) and Adv. Sci. **2201753**, 1 (2022), the different conductance states are associated with different oxidation states of the interface, that is, with oxygen migration into/from the interfacial YBCO. Because light produces the same effects (change of conductance) as electric pulses, it is natural to conclude that light is promoting oxygen migration into/from the interfacial YBCO. We discuss below how this happens, together with new supporting evidences.

Let us first discuss light effects in the ON state which, as shown in Fig. 2g, is metastable and naturally relaxes towards a lower conductance state at a rate that increases with temperature. Relaxation occurs

because oxygen migrates back into ITO (equivalently, oxygen vacancies move to YBCO) as dictated by electrochemical arguments, that is, due to the very different reduction potentials of both materials (see new supplementary Table 1), thus driving the system into its ground state –in which the interfacial YBCO is oxygen-depleted in favor of ITO oxidation (see new microscopy in Figs. 2b-c). The oxygen migration requires activation over the barrier for ion diffusion. In the dark, this process is thermally activated, as detailed in the analysis added to the new supplementary information Fig. S4. Upon illumination, the relaxation is dramatically accelerated, as shown in the new supplementary Fig. S5 and its analysis. This analysis shows that, under illumination, the system relaxes as it would expectedly do at an effective temperature around 200 K higher. This suggests that light excites phonons by inelastic scattering processes, which activates ion diffusion from YBCO into ITO similarly to temperature. This is consistent with the fact that the illumination effects in the ON state do not show a marked wavelength dependence in the UV to near-infrared range (supplementary Fig. S6). It is worth noting that the described scenario of light-enhanced oxygen mobility has been put forward earlier to explain persistent photoconductivity in YBCO [see e.g. PRB 47, 9017 (1993)]. In summary, light effects in the ON state can be summarized as follows: oxygen moves from YBCO into ITO driven by electrochemistry, the migration dynamics being activated by both light absorption and by thermal excitations.

In the OFF state, the ITO electrode is oxygen-rich and the interfacial YBCO is severely oxygen-depleted. A key element to understand the persistent conductance enhancement observed *after* illumination is its correlation with the photovoltage observed *during* illumination in the OFF state. Actually, a causal relationship exists between the photovoltage and the conductance increase. This is documented in the new supplementary Figs. S7 and S8, which demonstrate that the conductance enhancement is observed only in the presence of a photovoltage and scales with its size. This strongly supports the proposed microscopic mechanism, which is based on the fact that the photovoltage reflects a hole accumulation at the YBCO side of the space charge layer. This accumulation changes the doping of the interfacial YBCO, locally enhancing the number of holes per copper atom and thus the oxidation state $\text{Cu}^{+2} + \text{h}^+ \rightarrow \text{Cu}^{+3}$ beyond the level corresponding to the actual oxygen content. This favors the re-oxygenation of the interfacial YBCO by migration of O^{2-} ions. O^{2-} ions can migrate from the oxygen-rich interfacial ITO, where electron accumulation (Fig. 5a) allows O^{2-} to be released without changing the In oxidation state ($\text{In}^{+3} + \text{e}^- \rightarrow \text{In}^{+2}$). Notice that, consistently, the O^{2-} migration from ITO is also favored by the V_{OC} polarity, which is the same required to trigger that migration by application of V_{write} pulses. In addition, we cannot discard that O^{2-} diffuse into the interfacial YBCO also from the deeper YBCO layers that present higher oxygen content, given the sharp oxygen content gradient created after the application of $V_{write} < 0$. In summary, the positive photovoltage drives oxygen ions toward the oxygen-depleted interfacial YBCO, similarly as an applied $V_{write} > 0$ does. It is worth noting that this redox photocatalytic scenario has been established earlier [Jour. Photochem. Photobiol. A 1, 1-35 (1997)] for semiconductor heterostructures with a space-charge region (here the YBCO/ITO p-n junction formed at the junction interface). Notice finally, as seen in Fig. S8b, that there is a marked wavelength dependence: the photovoltage (and caused conductance enhancement) are observed in the UV/violet and drastically vanishes in the visible to IR range. This wavelength at which photovoltages are observed in the range 365-400 nm (3.4-3 eV), i.e. for photon energies above the YBCO charge transfer gap. This is not surprising as metal-ligand charge transfer excitations into CuO_2 plane states at energy losses higher than 3 eV have been recently reported in oxygen-depleted YBCO [see Scientific Reports volume 4, 7017 (2014)]

Changes made:

- We have included the new experiments in the supplementary Figures S4, S5, S6, S7, and S8 cited above.
- We have completely rewritten the discussion on pages 8-10 to better explain the origin of the photo-induced effects, as well as included in the argumentation the new pieces of evidence supporting the interpretation of the observed effects.

Referee's comment # 5. "What significant role does the superconductor play? What if the superconductor YBCO is replaced by other conducting oxides, such as (La,Sr)MnO3 or SrRuO3? A detailed discussion might be required. "

Our response: Superconductivity is important for two reasons. The first one is that the opening of the superconducting gap produces a drastic drop of the low-bias tunnel conductance (inset in Fig. 1e and new Supplementary Fig. S2), which ultimately results in a strong increase of the resistance switching amplitude [see Further discussion in [Nat. Commun. 11, 658 (2020)]. The second is that demonstrating photo-memristive effects in a high-temperature superconductor-based junction paves the way towards novel neuromorphic superconducting electronics, a nascent area that combines neuromorphic architectures with quantum coherence (Josephson) effects [see e.g. Supercond. Sci. Technol. 35, 053001 (2022)], to which our work is game-changing by endowing optical responses.

Yet the referee is right that similar optical switching effects should be observed in other conducting oxides, provided that the junction is made with a second material with different reduction potential to promote oxygen exchange, and that a p-n junction is formed at the interface. This is the case, for example, of NdSrNiO₃ (NSNO) and NdNiO₃ (NNO), for which we have recently demonstrated electrical

switching effects [Adv. Sci. 2201753, 1 (2022)] which are analogous (*mutatis mutandis*) to those in YBCO. Following the referee's suggestion, we have fabricated NSNO/ITO and NNO/ITO junctions and performed experiments under optical excitation and found the behaviors observed for YBCO/ITO, namely a conductance decrease when illumination in the ON state (a), as well as a photovoltage (b) and a conductance increase in the OFF state (c), as shown in the figure R3.

Because these results are preliminary and NNO has specificities that change the temperature behavior as compared to YBCO [Adv. Sci. 2201753, 1 (2022)] further experiments are being carried out to fully understand the systems before publication. For this reason, we prefer not to include the above data in the Supplementary Information section.

Changes made:

- We have rewritten the concluding paragraph to explain that analogous behavior can be expected with other complex oxides.

Referee's comment # 5 *“Other minor points: The reference format is not consistent and the descriptions in many places are too long and should be simplified, especially for the figure caption.”*

Our response: We thank the referee for pointing out these ways of improvement. We have simplified the text as much as possible and revised the references list.

We hope that the referee finds the revised manuscript suitable for publication.

Response to reviewer #2

We thank the referee for their positive appraisal and for stating the reported effects are a “novel, good finding”. We also appreciate the suggestions and criticisms, which we think have helped us improve the paper via new experiments, supplementary data, and a substantial rewriting of the manuscript. A detailed point-by-point answer to the questions raised by the referee is given below, with indications of the changes made. We hope that the referee finds the revised version of the manuscript suitable for publication

Referee’s comment #1: *“I am not very convinced of the technological merits that the authors are compelled to mention. For any sort of merits in a neuromorphic architecture, one needs to be able to use these devices at room temperature, or else there is a lot of overhead required to cool these devices (at a circuit level). That will kill any potential benefits. The authors may instead argue for architecture in photon based quantum computing that would still require electrooptic effects, and low temperatures are OK there.”*

Our response: The referee is right in the sense that we do not think (nor claim) that the superconducting properties of the photo-memristor will be exploited in consumer electronics, for example. We realize we did not sufficiently detail in which way and for what applications its technological merits apply. However, we do think that the benefits of superconductivity are worth the cost of cooling down in some important cases, including in the areas of electronics and neuromorphic computing, in which superconductivity brings performances (or allows for specific functions) that can hardly be obtained otherwise, mainly by exploiting quantum phase coherence effects (Josephson physics). In these cases, the cost/benefit balance is further exacerbated by the high-temperature superconductors used here, because cooling down to liquid nitrogen temperatures is relatively simple and inexpensive (for example, can-of-soda size refrigerators developed in our lab allow cooling superconducting circuits to 50 K in about 2 minutes, with a ~7 W consumption, less than a computer’s CPU and comparable to the ~3 W consumed by the fan that keeps it from overheating). This has given enormous momentum to the industrial development of high- T_c superconducting applications, particularly in quantum sensing, medical imaging, and defense areas (“conventional superconducting electronics”). Of course, comparatively, superconducting neuromorphic computing is still in its infancy. Yet it has much potential (due to high speed and the possibility of merging neural and synaptic functions [see e.g. Supercond. Sci. Technol. 35 053001 (2022)]), and we think the photo-memristive behavior demonstrated in our work adds an interesting ingredient to that effort.

Changes made:

- We have rewritten the concluding paragraph to more precisely put in context the technological potential of the effects demonstrated in the manuscript.

Referee’s comment #2: *“I am not quite sure what is the use of YBCO here. Can this photomemristive phenomena not work with any oxide that undergoes topotactic reactions (LSMO, for e.g.). “*

Our response: As the referee suggests, similar optical switching effects should be observed in other oxides, provided the junction is made with a second material with different reduction potential to promote oxygen exchange, and a p-n junction is formed at the interface. This is the case, for example, of NdSrNiO₃ (NSNO) and NdNiO₃ (NNO), for which we have recently demonstrated electrical switching effects [Adv. Sci. 2201753, 1 (2022)] which are analogous (*mutatis mutandis*) to those in YBCO. Following the referee's suggestion, we have fabricated NSNO/ITO and NNO/ITO junctions and performed experiments under optical excitation and observed all the behaviors observed for YBCO/ITO, namely a conductance decrease when illumination in the ON state (a), as well as a photovoltage (b) and a conductance increase in the OFF state (c), as shown in the figure R3.

Because these results are preliminary and NNO has specificities that change the temperature

behavior as compared to YBCO [Adv. Sci. 2201753, 1 (2022)], further experiments are being carried to fully understand the systems before publication. For this reason, we prefer not to include the above data in the Supplementary Information section.

Changes made:

- We have rewritten the concluding paragraph to explain that analogous behavior can be expected with other complex oxides.

Referee's comment #3: "Is the advantage of a superconductor only that ON-OFF ratios enhance below T_c? I couldn't clearly understand the importance of T_c in photoinduced effects."

Our response: Superconductivity is important for two reasons. The first one, as stated by the referee, is that the opening of the superconducting gap produces a drastic drop in the low-bias tunnel conductance (inset in Fig. 1e and new Supplementary Fig. S2), which ultimately results in a strong increase of the resistance switching amplitude [see Further discussion in Nat. Commun. 11, 658 (2020)]. The second is that demonstrating photo-memristive effects in a high-temperature superconductor-based device enhances their technological merits, as discussed above. However, as also discussed in response to query #2, superconductivity is not a necessary ingredient for the photoinduced effects.

Changes made:

To avoid confusion about the role of superconductivity, we have made the following changes:

- We have changed the title of the manuscript to "Bimodal ionic photo-memristor based on a high-temperature oxide superconductor/semiconductor junction".

- In the text, we explicitly state that superconductivity is not a necessary ingredient for the photomemristive effects (page 3, 2nd paragraph).

Referees comment #4: *“About figure 2d which shows the relaxation of the ON state, does it relax finally to an OFF state? Or does it asymptotically settle down at a value of $\Delta G_e(T)/\Delta G_{on}$, which is non-zero. Even at 250 C, I don't see this approaching 0. Which means that ON state is kinetically frozen for a long time at room temperature. That's OK from a device perspective.”*

Our response: Indeed, in the experimental conditions, the OFF state is not reached upon relaxation from the ON state at finite times. This is described by the stretched exponential behavior. We realize that this was not sufficiently discussed in the manuscript. The relaxation rate depends on temperature via a thermally-activated characteristic time scale, and also depends on illumination conditions as light absorption also activates the relaxation. This is illustrated via new supplementary figures S4 and S5, and in the related discussions, as we detail below. What these new figures convey is that, once the ON state is set electrically, the conductance level reached after a fixed relaxation time can be controlled by varying the temperature and/the illumination time and power.

Changes made:

- We have included new Fig. S4 and S5 to quantitatively analyze relaxation behavior in the dark and under illumination.
- The related discussions in the main text have been extended in page 5 (bottom paragraph) and page 8 (origin of optical switching effects).

Referee's comment #5: *“When ON state sees photons its resistance increases (non-volatile fashion). I am guessing once light is removed, this should also decay with time (just like it happens in the ON state under dark conditions). Please report that data, and comment on the retention. Same applies for OFF state. If the decrease in resistance is due to a photovoltaic effect, which is volatile, the so-called “non-volatile” increase in resistance should have a time scale associated with it, before which it recovers back to the starting OFF state. Please present retention data on both OFF and ON states after illumination is removed (at different read voltages).”*

Our response: The retention depends essentially on temperature. As the referee rightfully expects, after the junction is set into an intermediate conductance level by optical excitation and its removal, we observe a relaxation towards the OFF state. The relaxation rate depends on temperature, as one could anticipate from Fig. 2g and the new Supplementary Fig. S4. That can be seen in the new Fig. S9, which shows the experiments requested by the referee. At low temperatures, the intermediate, optically-induced conductance levels are virtually non-volatile. However, at 100 K one can already see a significant relaxation and, at higher temperatures, the junction can nearly reach the OFF state within 1h (provided the intermediate state previously set by illumination is not too far from the OFF state).

Changes made:

- We have included a new supplementary Fig. S9 that shows the measurements requested by the referee.
- We have included in the text a paragraph explaining the relaxation behavior after illumination, on page 6, 2nd paragraph.

Referee's comment #6: *"The idea of a pn junction to explain volatile PV effect is very interesting. However, the I-V transport shown in figure 4a doesn't seem to fit with the behavior of a diode. Even if the electron transport were more complicated than a simple diode, I expect some sort of asymmetry between negative and positive voltages owing to the presence of a pn junction. Rather these I-V curves look like SCLC type conduction (to speculate). Please comment, and prove that there is a pn junction through I-V measurements, and if necessary simple circuit modeling."*

Our response: The I-V curves look symmetrical at low-bias because this is the regime in which the conduction is dominated by electron tunneling, as discussed in the manuscript and supported by the new supplementary Fig. S1 (which shows fits of the low-bias conductance to the Brinkman-Dynes-Rowell model for electron tunneling) and Fig. S2 (which shows the behavior expected from electron tunneling into a superconductor). However, the asymmetry characteristic of the pn junction is evident in the new supplementary Figure S10, which shows data for an extended bias range in which the asymmetry characteristic of a pn junction is evident.

Changes made:

- We have included a new supplementary Fig. S10 that shows the I(V) measurement displaying the asymmetry typical of p-n behavior, as requested by the referee. This is recalled in the main text where the pn scenario is brought up (page 8, last paragraph).

Referee's comment #7: *"For the ON state resistance to decay with photons, what exact redox reactions are important for this. Please detail the chemistry. And does any wavelength of light work? Or should one be in a wavelength where carriers are generated (UV)? In other words, infrared radiation may not do anything (if heating is excluded). Furthermore, the suggestion that photons somehow enhance diffusion (without heating), would mean momentum transfer from photons to oxygen vacancies. Photon momentum is negligible, that photomigration is a difficult process to imagine. If this is not the case, what do photons do to enhance the ion migration?"*

Our response: The relaxation from the ON into the OFF state is produced by a redox reaction in which Cu in YBCO is reduced (it has the highest reduction potential $E_0=2.4$ V) and In in ITO oxidizes ($E_0=-0.49$ V) :

This redox reaction is realized as O^{2-} ions move from YBCO into ITO leaving oxygen vacancies in the interfacial YBCO. It is spontaneous from the electrochemical point of view, and results in the system's ground state, as shown by the new microscopy and spectroscopy experiments shown in Figs. 2b-2d. When the junction is set in the ON state, the interfacial YBCO is oxygen-rich and ITO oxygen deficient, and the junction is therefore out of equilibrium from the electrochemical point of view. This explains the natural decay into the OFF state (ground state). However, for the above reaction to occur, the barrier diffusion for O^{2-} needs to be overcome. As shown in Fig. 2g and the new supplementary Fig. S4, ion diffusion is thermally activated. Light absorption also provides energy to activate ion diffusion, further accelerating the decay of the conductance, as shown in the new Supplementary Fig. S5. That is, light affects the dynamics of the redox reaction. Indeed, as demonstrated by the analysis in Fig. S5, relaxation under light illumination is as fast as if the temperature was 200 K above that measured by using as thermometer the YBCO electrode itself. This can be understood considering that light absorption (e.g. due to inelastic scattering) excites lattice vibrations (Raman effect). This is consistent with the fact that the photo-induced acceleration of the relaxation does not show a marked

wavelength dependence. This is shown in the new Supplementary Figures S6 and S7: all the studied wavelengths within the visible -near IR range produce strong effects, including IR, which indeed produces very strong effects.

Changes made related to the above:

- We have included a new Supplementary Figure S5 to quantitatively analyze how light accelerates the dynamics of the spontaneous relaxation from the ON into the OFF state.
- We have significantly rewritten the manuscript to describe the naturally occurring redox reaction between YBCO and ITO and included microscopy experiments (new Figs. 2b-c-d) that show the interfacial ground state (page 4).
- We have included a new Supplementary table with the reduction potential to support the electrochemical arguments.
- The redox reactions and how they are reversed by the different light effects are explicitly included in the discussion of the optically induced switching (page 8).
- We have performed and included new experiments as a function of the wavelength, shown in the new Supplementary Figs. S6, S7 and S8.

Referee's comment #8: *"Since the ON state increases in resistance with illumination, OFF state decreases with illumination, if shone long enough on the ON state say, the increase in resistance should slowly subside and there should an upturn after which resistance decreases. Do the authors see this? What are the typical time scales? If that is not the case then ON state is not really relaxing to an OFF state, but rather relaxing to a different state."*

Our response: The conductance decrease observed upon illumination in the ON state does slowly subside, as suggested by the referee. However, it is not followed by an upturn. The referee is right in his/her second guess, namely, that under illumination the system does not reach the OFF state but, rather an intermediate state between ON and OFF. We regret this was not better explained in the earlier version. The light effects in the ON and OFF states are antagonistic. To understand how they balance, it is important to recall, as we show in the new Fig. S8a, that the photoinduced conductance increase observed with UV light in the OFF state is correlated with the measured photovoltage V_{oc} . The photovoltage is absent in the ON state but progressively emerges as the system is driven into the OFF state (see Fig. 4b). Thus, one can see that when the system is illuminated in the ON state, the conductance gradually decays (as it would naturally do in the dark, although faster due to light effects). However, as the relaxation goes on, the photovoltage V_{oc} gradually grows and drives the junction's conductance in the opposite direction (due to photodoping at both sides of the junction, as explained in the revised text, page 9, 2nd paragraph), which at some point stops the relaxation. That is, at some point the two antagonistic effects balance. Consequently, the junction can't be switched all the way from the OFF into the ON state (nor vice versa) by illumination in the UV range, where both effects exit, but it is rather driven into an intermediate conductance level. Whether this level is closer or farther from the OFF state will depend on temperature, as thermal excitation also pushes the junction towards the OFF state, tipping the balance between the antagonistic photoinduced mechanism. Notice that the junction conductance decay will stop at some intermediate level under illumination, but once in the dark, it will follow its natural towards the OFF state at a rate that depends on temperature, as demonstrated in the new supplementary Fig. S9.

Changes made:

- In the revised version of the main text, we clearly state that, upon illumination, it is not possible to optically switch the junction from ON to OFF or vice versa, but instead that the junction conductance evolves towards an intermediate level between ON and OFF (page 6, 2nd paragraph).
- We have rewritten the concluding paragraphs of the discussion section (particularly in the last starting on page 9) to explain why an intermediate state is reached, due to the interplay between the competing effects.

Referee's comment #9: *"For photon integration or photon counting, one needs to show how the detected parameter (ΔG) changes with the photon intensity. If this is linear over a large range, it is a good detector. Here we don't know what the characteristics of ΔG vs photon power are. If the authors are to claim that this will be good photon counter, they must first show this relation. ΔG may saturate at larger powers, and that may not be that useful."*

Our response: We thank the referee for pointing out that this was not clear. That the conductance changes with the photon intensity is demonstrated in Fig. 3c. That figure shows that the size of the photoinduced conductance changes correlate with the illumination time (if power is kept constant) and with the power at a fixed time. Indeed, the response is not linear (notice that the graph is in a double log scale), and will saturate at some point (see the response to question 8). Therefore as suggested by the referee this may not be an ideal photon counter. But we think that the data at hand allows for the claim that the devices behave as a photon integrator (the conductance change scales with the number of photons shone on the device) within the experimental window (e.g over almost two time decades). To stress the cumulative effect, we have included in a new Supplemental Figure S11 that shows conductance curves after illumination with different powers.

Changes made:

- We have stressed in the description of Figure 3c that the response is not linear.
- We have included a new supplementary Figure S11 with shows raw data to illustrate the relationship between illumination power and induced conductance variation.

Response to Referee #3:

We thank the referee for their positive appraisal of the manuscript and for the constructive criticism, which was in essence that the discussed mechanisms were not sufficiently substantiated in the earlier version. Following this criticism, we have performed new experiments, some of them as proposed by this referee, as well as included 11 new supplementary figures, and substantially rewritten the manuscript. We hope that the referee finds the revised version suitable for publication.

Referee comment #1: *“This manuscript deals with an interesting concept for memristive devices. The authors show that the interface between a bottom YBa₂Cu₃O₇ (YBCO) electrode and an indium-tin-oxide (ITO) top electrode can behave as a tunnelling photo-memristor. The authors argue that the physical properties of the oxide heterointerfaces are determined by electronic and/or ionic reconstructions. More interesting, both electrical and optical stimuli are shown to be able to alter the resistance states, depending on both optical and/or electrical history. Such light sensitivity to electrical memristors might open possibilities for complex computational applications involving both electrical and optical switching in a single passive component. Although the device shows clear bimodal memristive behavior, and the concept might be considered novel, ...”*

Our response: We are glad that the referee finds that our manuscript reports on an “interesting concept for memristive devices”, and that “the bimodal memristive concept can be considered novel”.

Referee comment #2: *“I feel that the manuscript lacks detailed understanding of the behavior. The authors assume that the physical properties of the oxide heterointerfaces are determined by electronic and/or ionic reconstructions. Mainly hand-waiving arguments have been described in the manuscript. In general, both the measured effects and the suggested mechanisms are described sufficiently. However, the authors did not provide evidence for these mechanisms.*

Our response: We are glad that the referee finds the description of the effect and suggested mechanism sufficiently clear. To substantiate the discussed mechanisms, we have included in the manuscript new supporting data, experiments, and analysis as detailed in what follows.

Changes made:

1) To support electron tunneling as the conduction mechanism in the low-bias regime,

-We have included a new supplementary Figure S1 that shows typical fits of the conductance vs. bias to the BDR model, which can be made in the normal state of YBCO. These fits allow for estimating the tunnel junctions' parameters (barrier thickness and height) that are reported in the main text.

- We have included a new supplementary Figure S2 that displays a set of conductance vs. bias curves at different temperatures. As detailed in the figure caption, that set of data behaves as expected for electron tunneling into a superconductor, which supports electron tunneling as the governing tunneling mechanism in the studied junctions.

2) To demonstrate that oxygen exchange across the interface causes the resistance switching:

- We have performed microscopy (STEM) and spectroscopy (EELS) of the YBCO/ITO interface (new Fig. 2b), which demonstrates a change of the oxidation state of Cu in YBCO within a few nanometers from the interface, supporting oxygen migration into ITO as well as

the scenario sketched in Fig. 2a to explain the resistive switching effects. This new data are shown in new Figs. 2b, 2c & 2d.

- We have included a new supplementary Fig. S3 that shows that the large resistance switching effects are obtained only if YBCO is interfaced with a material that has a higher tendency to oxidize. We have rewritten the text to accommodate the information above as well as to point at our earlier publications in similar systems where this is detailed [Nat. Commun. 11, 658 (2020)] and [Adv. Sci. 2201753, 1 (2022)]. The rewritten text is found between pages 4 and 5.

3) To support that oxygen migration from YBCO into ITO causes the photo-induced conductance decrease upon illumination in the ON state:

- We have included the new experiments in the supplementary Figure S4, which allow insight into the natural relaxation from the ON to the OFF state (conductance decrease) that occurs in the dark. This figure and its analysis demonstrate that the conductance decay occurs naturally because this drives the system into its ground state, and also that the process is thermally activated.

- We have included new experiments to qualitatively analyze how illumination makes the above relaxation faster, particularly that under illumination the relaxation rate is as the temperature was 200 K above the measured temperature. This is shown in the new Supplementary Figure S5.

- We have included new experiments as a function of the illumination wavelength (supplementary Figures S5 & S6) that show that, in the ON state, illumination produces similar effects for all the wavelengths in the visible-near IR range. This supports that light absorption (Raman effect) provides energy to activate ion diffusion.

- We have completely rewritten the discussion on these effects (pages 8 and 9) and included references to earlier work that considers and discusses photoinduced ion diffusion in the cuprates [see e.g. PRB 47, 9017 (1993)]

4) To support that the interface behaves as a p-n junction in the OFF state and that this explains the photovoltaic effects

- We have included in Supplementary figure S10 an example of I(V) in an extended bias range, which shows the behavior expected for a p-n junction.

- We have included a new experiment of the photovoltage as a function of the wavelength (new supplementary Fig. S7 and S8b). This shows that these effects are only strong in the UV/violet and drastically vanish in the visible to IR range. This wavelength at which photovoltages are observed in the range 365-400nm (3.4-3 eV), i.e. for photon energies above the YBCO charge transfer gap. This is not surprising as metal-ligand charge transfer excitations into CuO₂ plane states at energy losses higher than 3 eV have been recently reported in oxygen-depleted YBCO [see Scientific Reports 4, 7017 (2014)]

5) To support that oxygen migration from ITO into YBCO causes the photo-induced conductance increase upon illumination in the OFF state:

- We have included a new supplementary Fig. S8 that demonstrates a causal relationship between the photovoltage and the conductance increase.

- We have completely rewritten the related discussion on pages 9 and 10 to better explain the origin of this photo-induced effect, as well as included in the argumentation the new pieces of evidence supporting the interpretation of the observed effects and references [see. E.g. Jour. Photochem. Photobiol. A 1, 1-35 (1997)] where the suggested mechanism is theoretically described.

Referee comment #3: “Furthermore, the crystalline quality and the interface characteristics (such as roughness) are very much dependent on the synthesis routes and reproduction of these devices might be hard in other labs. **More info on materials characteristics of the interface might be necessary.** To warrant publication in Nature communications, I feel that the manuscript needs to be significantly improved”.

Our response: Following this referee’s criticism, we have carried out a careful structural and chemical characterization of the junction’s interface, in particular via Scanning Electron Microscopy (STEM) and Electron Energy Loss Spectroscopy (EELS). These experiments have allowed us, on the one hand, to prove the high-structural quality of the interfaces, and on the other hand, to determine the oxidation profile near the interface. This indeed provides strong evidence of the oxygen exchange between YBCO and ITO. We think that these experiments directly address the main specific criticism raised by the referee.

Changes made:

- The new experiments are shown in Fig.2b,2c, and 2d, and are discussed in detail in the dedicated paragraph on page 4

Referee comment #4: *“Interfaces in heterostructure with YBCO (in contact with for instance metals or other oxides) have been studied a lot in the last decades, and redox mechanisms have been shown to be responsible for altering the properties of the interface. Several other experimental results, for instance from spectroscopic and transport measurements, indicate the role of redox driven reactions. As a general feature for cuprate superconductors, the Cu-O bond is weak, and redox reactions easily occur at the cuprate interface with the contact layer. Such reactions can result in electron doping, oxygen (vacancy) migration, complete destruction of superconductivity in the interface region, or modification of the YBCO microstructure. Next, electric field manipulation have been shown as an effective method to tune the redox-induced effects, and these experiments show that electrochemical modification of interfaces results in a nontrivial spatial profile of the oxygen vacancy distribution close to the interface. Furthermore, optical modulation of the resistance of memristors is not new and photomemristors and their behavior, such as MoS₂ nanosphere-based and Al/perovskite” FAPbBr₃/ITO heterostructures, have been demonstrated before. Also in such structures, modification of the resistance state is due to ionic and/or electronic effects.”*

Our response: We are glad that the referee acknowledges that the effects reported here are an “interesting concept for memristive devices” and that “the bimodal memristive concept can be considered novel”. Indeed, the key novelty of our work is that we show a coupled memristive (electrical) and photo-memristive (optical) behavior. Likewise, we are glad that the referee acknowledges that the main mechanisms we suggest to explain those effects (namely oxygen exchange between YBCO and ITO, its effect on the interfacial physical properties, and that it can be triggered by electric-fields and light) are plausible based on the literature.

Changes made:

- Following the referee's comment, we have included some references on the related effects and systems (New Refs 30,31,52 & 60)-

Referee comment #5: *"In conclusion, I feel that the authors show an interesting device, and the optical/electronic resistive switching behavior is well-described. The manuscript however lacks materials science aspects of the device and sufficient proof for the suggested mechanisms. I, therefore, propose to reject this manuscript for publication in nature communications.*

Our response: We are glad that the referee acknowledges the interest in the investigated devices, and that the novel behaviors are well described. We hope that they will find that the substantial amount of new data included in the revised version supports the suggested mechanisms, and also that the main criticism of the lack the characterization of the interfaces is addressed with the thorough microscopy and spectroscopy experiments we provide. We hope that this will allow the referee to support the publication of the revised version in Nature Communications.

REVIEWER COMMENTS

Reviewer #1 (Remarks to the Author):

The manuscript reported a resistive switching effect at a superconductor/semiconductor junction, where the conduction on-off ratio can be modulated by both electric field and photo-illumination. The authors claim that the microscopic mechanism at play is a reversible nanoscale redox reaction between both materials, whose oxygen content determines the electron tunnelling rate across their interface. Oxygen exchange is controlled here via illumination by exploiting a competition between electrochemistry, photovoltaic effects and photo-assisted ion migration. My concern is the mechanism of oxygen exchange. I disagree the statement, "the competition between electrochemistry, photovoltaic effects, and photo-assisted ion migration." It is too general, the authors should discuss the specific mechanism of the oxygen exchange. Normally, the oxygen migration is low, both electric field and photo could assist the migration, which one is key role in determining the migration. In addition, can the authors calculate/measure/estimate the concentration of oxygen or ion?

Reviewer #2 (Remarks to the Author):

I am overall happy with the response to my questions. I recommend that the paper be published, with editorial checks in language and formatting.

Reviewer #3 (Remarks to the Author):

My main criticism was the lack of detailed info on the materials science as well as the main mechanisms. The authors did a great job in the response, the additional studies and the changes in the manuscript. From my point of view, the manuscript fulfils the requirements for publication.

RESPONSE TO REVIEWER 1

We thank the referee for his/her review. A detailed point-by-point response is given below

Referee comment # 1 *“The manuscript reported a resistive switching effect at a superconductor/semiconductor junction, where the conduction on-off ratio can be modulated by both electric field and photo-illumination. The authors claim that the microscopic mechanism at play is a reversible nanoscale redox reaction between both materials, whose oxygen content determines the electron tunnelling rate across their interface. Oxygen exchange is controlled here via illumination by exploiting a competition between electrochemistry, photovoltaic effects and photo-assisted ion migration”*

Our response: We thank the referee for summarizing some of the paper’s findings.

Referee comment # 2 *“My concern is the mechanism of oxygen exchange. I disagree the statement, “the competition between electrochemistry, photovoltaic effects, and photo-assisted ion migration.” It is too general, the authors should discuss the specific mechanism of the oxygen exchange”*

Our response: That statement is an excerpt from the abstract, where, due to space limitations, the mechanisms at play can only be enumerated. We regret that the Referee may have found it confusing. We have revised and reformulated it (see below). However, a detailed discussion on how the enumerated mechanisms control oxygen exchange, and thus the observed photo-memristive effects, can be found in the manuscript. That discussion spans from page 8 to page 10, and is accompanied by Fig. 5. There we show that there is not only “one” mechanism involved, as the referee seems to expect. Oxygen exchange is controlled by the interplay between the three mechanisms. Oxygen diffusion, driven by the difference in the reduction potentials, is spontaneous as demonstrated by the relaxation phenomenon. It can be further promoted by illumination as demonstrated by the change in the relaxation rate under illumination. And electric field (either applied externally or produced by the photovoltage) can tip or invert the oxygen diffusion trend, thus allowing the bidirectional change in the resistance states.

Changes made: The statement in question has been rewritten to emphasize that there is an interplay between different mechanisms (sentence marked in yellow in the revised abstract) . The new sentence reads *“The redox reaction is optically controlled via an interplay between electrochemistry, photovoltaic effects and photo-assisted ion migration”*

Referee comment # 3 *“Normally, the oxygen migration is low, both electric field and photo could assist the migration, which one is key role in determining the migration.”*

Our response: We are glad that the referee agrees that both electric field and photo-excitation can assist oxygen migration. However, as discussed in detail on pages 8 to 10, there is no unique answer to the question “which is the key mechanism in determining the migration”: one or the other can be dominant, depending on the state of the junctions (ON/OFF) and the presence or not of an externally applied electric field.

Referee comment # 4 *“In addition, can the authors calculate/measure/estimate the concentration of oxygen or ion?”*

Our response: The oxygen content cannot be measured in absolute terms, as here this is a nanoscale, local variable. However, and crucially, its variation as a function of the distance from the interface can be inferred from the microscopy and spectroscopy experiments reported in Fig. 2b-d. This figure, added in the previous revision of the manuscript, shows that the YBCO’s oxygen content is lower near

the interface with ITO and reaches a constant value ~ 5 nm from the interface, which implies oxygen migration from the interfacial YBCO. This is discussed in detail on page 4.